# Machine learning identifies key metabolic reactions in bacterial growth on different carbon sources

Hyunjae Woo (ID), Youngshin Kim (ID), Dohyeon Kim & Sung Ho Yoon (ID) ✉

## Abstract

**Carbon source-dependent control of bacterial growth is fundamental to bacterial physiology and survival. However, pinpointing the metabolic steps important for cell growth is challenging due to the complexity of cellular networks. Here, the elastic net model and multilayer perception model that integrated genome-wide gene-deletion data and simulated flux distributions were constructed to identify metabolic reactions beneficial or detrimental to *Escherichia coli* grown on 30 different carbon sources. Both models outperformed traditional in silico methods by identifying not just essential reactions but also nonessential ones that promote growth. They successfully predicted metabolic reactions beneficial to cell growth, with high convergence between the models. The models revealed that biosynthetic pathways generally promote growth across various carbon sources, whereas the impact of energy-generating pathways varies with the carbon source. Intriguing predictions were experimentally validated for findings beyond experimental training data and the impact of various carbon sources on the glyoxylate shunt, pyruvate dehydrogenase reaction, and redundant purine biosynthesis reactions. These highlight the practical significance and predictive power of the models for understanding and engineering microbial metabolism.**

**Keywords** Machine Learning; Deep Learning; Carbon Source; Bacterial Growth; Metabolic Reaction
**Subject Categories** Computational Biology; Metabolism

## Introduction

Bacteria adjust their metabolic pathways according to nutrient availability (Litsios et al, 2018; Shimizu, 2016). The identification of metabolic genes and pathways that promote or retard cell growth is crucial for understanding how bacteria modulate cellular metabolism to efficiently utilize available carbon sources. However, the identification of growth-controlling genes is challenging because of the robustness and complexity of cellular networks. The regulation of bacterial growth involves a complex interplay among cellular composition, ribosome concentration, and metabolism (Jin et al, 2012), which makes it difficult to identify the specific genes and pathways that directly control growth.

Various experimental and computational approaches have been used to explore the contribution of different metabolic processes to bacterial growth. High-throughput genetic screening methods, such as transposon mutagenesis or gene-deletion libraries, have been employed to systematically investigate the impact of individual gene disruptions on bacterial growth (Tong et al, 2020). Transcriptomics, proteomics, and metabolomics have also been instrumental in identifying growth-controlling genes (Jiang et al, 2015). By comparing the expression profiles under different growth and physicochemical conditions, researchers can identify genes and metabolic pathways that are differentially expressed or modulated. In addition to experimental techniques, mathematical modeling has emerged as a powerful tool for understanding and predicting the behavior of complex biological systems (Gu et al, 2019; O'Brien et al, 2015). Computational models have been used to predict the effects of genetic perturbations on growth (Joyce and Palsson, 2008; Long and Antoniewicz, 2014), generating testable hypotheses that can guide further experimental investigations.

The integration of experimental and computational approaches can offer a more accurate identification of key metabolic genes and pathways that influence bacterial growth under specific environmental conditions. Machine learning (ML) and its specialization, deep learning (DL), have become pivotal tools for extracting useful patterns and relationships from complex, multi-dimensional, and large-scale biological and biomedical data (Inza et al, 2010; Larrañaga et al, 2006; Min et al, 2017). However, many ML approaches, particularly DL, have been criticized for lack of interpretability of their outcomes, which is called the "black-box" problem, causing ambiguity in how they work and how decisions are made (Linardatos et al, 2020; Loyola-González, 2019). To overcome this limitation, constraint-based metabolic modeling (CBM) has been combined with ML to provide mechanistic insights into genotype-phenotype relationships (Antonakoudis et al, 2020; Sahu et al, 2021; Zampieri et al, 2019). CBM-based metabolic flux profiles can be used as input data to train ML models with experimental screening measurements as the output data. Specifically, this approach has been employed to investigate the biological mechanisms underlying antibiotic lethality (Yang et al, 2019) and to predict the growth rate of yeast (Culley et al, 2020). These studies highlight the enhanced predictive capacity of combining CBM and ML, and emphasize their potential to uncover the causal mechanisms behind complex phenotypes.

Department of Bioscience and Biotechnology, Konkuk University, Seoul 05029, Republic of Korea. ✉E-mail: syoon@konkuk.ac.kr

Establishing a growth control strategy is important for microbial bioprocess development. *E. coli* is the preferred platform for numerous industrial applications owing to its fast growth, ability to achieve a high cell density, well-established genetic tools, and the capability to overexpress recombinant proteins (Hayashi et al, 2006; Rosano and Ceccarelli, 2014; Yoon et al, 2009). *E. coli* K-12 is the best-studied model microbe for which a comprehensive single-gene-deletion mutant (Keio) collection (Baba et al, 2006) is available. In this study, we identified metabolic steps that are beneficial or detrimental to *E. coli* K-12 grown on 30 different carbon sources. Experimental growths of single-gene deletions and metabolic simulations were combined to train two explainable ML models. Predictions from both models were evaluated using experimental growth data and combined to identify the metabolic reactions that significantly influenced bacterial cell growth with different carbon sources. Furthermore, intriguing predictions were tested by performing growth experiments on the corresponding gene-deletion mutants.

# Results

## Workflow of this study

We generated predictive computational models that provide mechanistic insights into *E. coli* growth on different carbon sources. To identify the metabolic responses affecting cell growth, predictive models were constructed using supervised learning algorithms that mapped metabolic fluxes derived from minimization of metabolic adjustment (MOMA) simulations as input data to experimental growth data as output data (Fig. 1A). In general, simple conventional ML methods, such as linear regression and decision trees, provide high interpretability with compromised prediction accuracy. In contrast, complex DL methods have a high prediction accuracy for the black-box problem (Linardatos et al, 2020). Considering the weaknesses of each approach, we generated two predictive models: an ML-based elastic net (EN) regression model (Zou and Hastie, 2005) and a DL-based multilayer perceptron (MLP) model (Gardner and Dorling, 1998) (Fig. 1B). The predictive accuracies of the constructed models were evaluated (Fig. 1C). Subsequently, we combined the predictions from both models and performed functional enrichment analysis to identify overrepresented metabolic pathways within the predictions. The new biological insights and discoveries predicted by the models were experimentally validated to confirm the predicted effects of specific metabolic reactions or carbon sources on cell growth (Fig. 1D).

## Preparation of training dataset for supervised learning

Supervised learning requires both input and output data. Growth data were obtained from previous studies (Tong et al, 2020) that measured the cell growth of gene-deletion mutants for 3796 genes of *E. coli* K-12 BW25113 (Keio collection) (Baba et al, 2006) growing on solid agar containing minimal media with 30 different carbon sources for 24 h. Of the 3796 genes, 1295 were associated with the *E. coli* K-12 metabolic network model (iML1515) (Monk et al, 2017) and the 24 h endpoint biomasses of the corresponding mutants were used as the output data. In addition to the 1295

genes, the endpoint biomasses associated with the deletions of 127 metabolic genes that were experimentally determined to be essential (Baba et al, 2006) were set to zero. Thus, the output dataset was formulated as a $1422 \times 30$ matrix with 1422 biomasses of gene-deletion mutants under 30 different carbon sources (Fig. 1A).

To generate the input data, we performed gene-deletion simulations using the genome-scale metabolic model (GEM) of *E. coli* K-12 MG1655 (iML1515) (Monk et al, 2017) and in silico MOPS minimal media supplemented with each of the 30 carbon sources (Tong et al, 2020). To simulate gene-deletion, metabolic reaction(s) associated with each of the 1422 genes tested in the output data were constrained to zero. MOMA was used to simulate the metabolic flux distribution of the mutant strain. MOMA is an algorithm specifically designed to predict the behavior of a mutant strain by minimizing changes in metabolic flux between the wild-type and mutant strain (Segrè et al, 2002). We also evaluated a different type of flux simulation methods such as flux variability analysis (Gudmundsson and Thiele, 2010) and flux sampling (Herrmann et al, 2019), which can generate non-zero flux values for a wide range of reactions in Appendix Text S1 and Appendix Fig. S1.

For each of the 30 carbon sources, the initial input data were formulated as a $1422 \times 2715$ matrix $X = [x_{ij}]$, where 1422 is the number of deletion mutants (observations), 2715 is the number of metabolic reactions (features), and $x_{ij}$ is the value of reaction flux $j$ in deletion mutant $i$. We took the absolute value of each flux to prevent the misinterpretation of negative flux values as representing low reaction activity rather than reverse reactions, during the model training. Then, we removed features that had zero values across all 1422 mutation simulations to reduce the high dimensionality of the data. This can be seen as a form of variance thresholding (Fida et al, 2021) where the threshold is set to zero, since all fluxes with constant values had a value of zero, except for only one reaction of the ATP maintenance requirement. We also evaluated two widely used techniques for feature selection, variance thresholding and random forest (Fida et al, 2021), in Appendix Text S2 and Appendix Table S1.

## Construction of the predictive models

Regarding the interpretability of the model and overfitting avoidance, regularized linear regression methods are widely used for model fitting and feature selection in the analysis of high-throughput biological data (Ajana et al, 2019). Particularly in cases where the number of features ($n$) is much larger than the number of observations ($p$), the EN is a preferred choice over lasso regression because it addresses the over-regularization problem by combining the properties of lasso and ridge regularization (Zou and Hastie, 2005). This situation ($n \gg p$) corresponded to the datasets used in this study, where the number of metabolic reactions ($n$) was approximately twice the number of deletion mutants ($p$). In the EN model, corresponding to each of the 30 different carbon sources, metabolic reactions with positive (or negative) correlations were considered beneficial (or detrimental) for cell growth under the given carbon conditions (Fig. 1B; Dataset EV1).

The MLP, a feedforward artificial neural network, can model highly nonlinear function mapping between an input layer and an output layer using a series of fully connected hidden layers

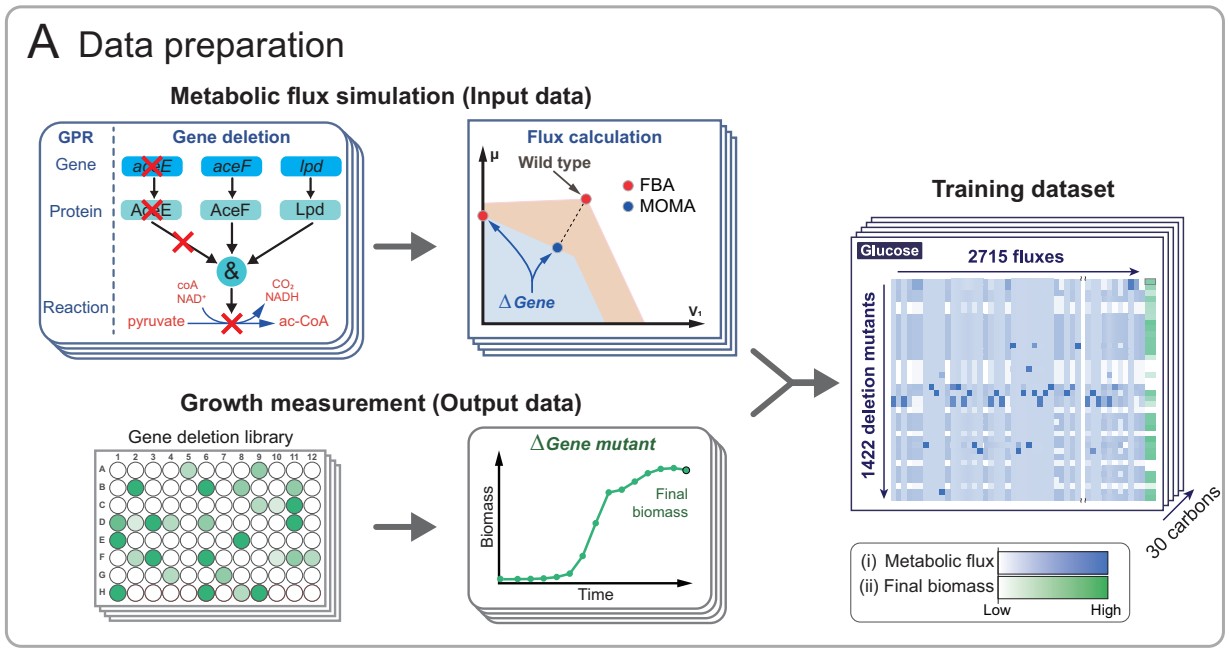

## A  Data preparation

**Metabolic flux simulation (Input data)**

**Growth measurement (Output data)**

**Training dataset**

(i) Metabolic flux
(ii) Final biomass
Low    High

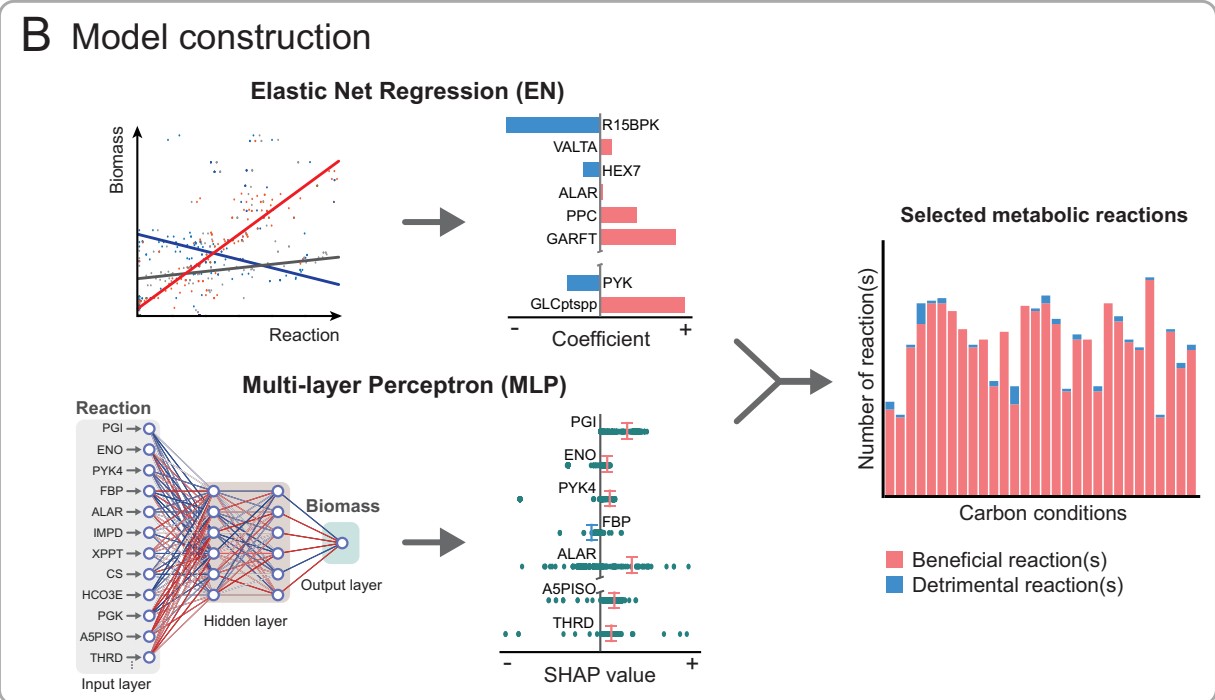

## B  Model construction

**Elastic Net Regression (EN)**

**Multi-layer Perceptron (MLP)**

**Selected metabolic reactions**

Beneficial reaction(s)
Detrimental reaction(s)

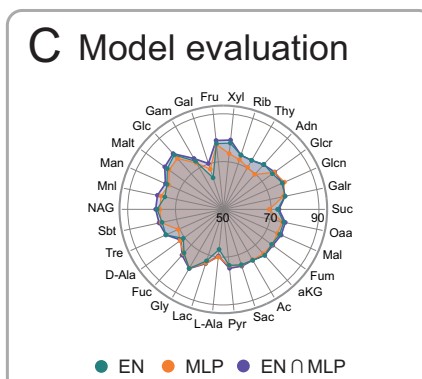

## C  Model evaluation

EN    MLP    EN ∩ MLP

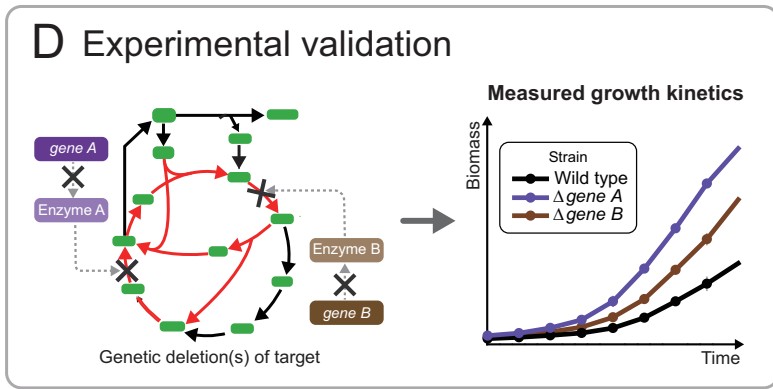

## D  Experimental validation

**Measured growth kinetics**

Genetic deletion(s) of target

◀ **Figure 1.  Explainable machine learning approaches to identify metabolic reactions significantly affecting bacterial growth under different carbon source conditions.**

(**A**) Preparation of training dataset for supervised learning. For output data, 1422 biomass measurements of a genome-wide gene-deletion library of *Escherichia coli* grown on 30 different carbon sources were collected from previous studies. For input data, metabolic flux distributions of gene-deletion mutants contained in the output data were simulated using metabolic simulation of minimization of metabolic adjustment (MOMA). (**B**) Construction of the predictive models. Two machine learning models, the elastic net (EN) regression and multilayer perceptron (MLP), were constructed using the training dataset. The relationships between qualitative input and qualitative output were learned to identify which input feature reactions have significant relationships with the output data. Metabolic reactions beneficial or detrimental to cell growth were identified based on the coefficient and SHAP value of each metabolic reaction, respectively, in the EN and MLP models. (**C**) Measurement of model accuracy. Model predictions were compared with experimental gene-deletion datasets. (**D**) Experimental validation of the model predictions. The predictions were validated in growth experiments using the gene-deletion mutants.

(Gardner and Dorling, 1998). We constructed an MLP model using the metabolic flux values as the input layer and the normalized growth data as the output layer. To overcome the lack of interpretability of the constructed MLP model, we employed the SHapley Additive exPlanations (SHAP) method, a unified framework based on game theory that assigns importance values to each feature or input variable in a prediction, thereby providing insights into the decision-making process of the model (Lundberg and Lee, 2017). For each metabolic reaction, the SHAP value was calculated to represent the expected marginal contribution over all possible feature combinations (Fig. 1B; Dataset EV1). The MLP model for each carbon condition was trained using a predetermined set of hyperparameters, and interpreted with SHAP across ten iterations. When training the model, both the training loss and validation loss converged to zero (https://github.com/sybirg/xai_growth/tree/main/Supplementary), indicating that the model is stable and the chosen hyperparameters were well-suited for this specific problem and dataset.

Both EN and MLP models had a strong tendency to predict beneficial metabolic reactions over detrimental reactions. On average, per carbon source, the MLP model predicted more beneficial reactions (353.2 ea) than the EN model (308.5) (Appendix Table S2). A significant majority (97% on average per carbon) of the beneficial reactions predicted by the EN model were also predicted by the MLP model, suggesting that these reactions were indeed likely beneficial. The MLP model often predicted the same beneficial reactions across various carbon source conditions, which resulted in fewer beneficial reactions (525 ea) than the EN model (565), with 405 common reactions, under at least one of 30 carbon conditions (Fig. 2A).

The disparity between the two models was pronounced when predicting detrimental reactions. The EN model predicted a considerably higher number of detrimental reactions (48.8 each per carbon on average) compared to the MLP model (17.5). Furthermore, only 3% of the detrimental reactions predicted by the EN model aligned with the MLP predictions, indicating substantial variability in how each model identified or classified detrimental reactions. Because dozens of detrimental reactions were predicted differently for each carbon source condition, the numbers of these predictions were summed to 480 and 282 by the EN and MLP models, respectively, with only 21 in common.

## Model accuracy

It can be speculated that the removal of beneficial metabolic reactions reduces the final biomass or even prevents cell growth. Regarding two datasets for phenotypic screening of single-gene deletions: one being the training dataset (Baba et al, 2006; Tong

et al, 2020) and the other an independent dataset (Monk et al, 2017), critical reactions within these datasets were obtained from the respective papers that reported these datasets. The accuracy of each model was evaluated by counting the number of critical reactions found among the positive reactions predicted by the model (Fig. 2B; Dataset EV2; "Methods").

Regarding the critical reactions reported in the training data (Baba et al, 2006; Tong et al, 2020), the EN model showed an average accuracy of 74.9% per carbon source, ranging from 63.9% for the galactose condition (85 out of 133 beneficial predictions agreed with the experimental results) to 80.7% (96/119) for the glucose condition. The MLP showed an average accuracy of 74.2%, ranging from 67.7% (88/130) for the galactose condition to 78.7% (100/127) for the glucose condition. When the metabolic reactions predicted by both models were considered, the prediction accuracy increased slightly to an average of 75.9% compared with both individual models.

Model accuracy was further evaluated using additional independent gene-deletion experimental data that were not used for model construction (Monk et al, 2017). The data contained a list of genes essential for growth on minimal media, with 16 different carbon sources. For the 14 carbon conditions implemented in the models, metabolic reactions associated with essential genes (Monk et al, 2017) were used to calculate predictive accuracy. The average accuracies for the 14 carbon conditions were 82.9% and 81.8% for the EN and MLP models, respectively. When the beneficial reactions predicted by both models were considered, the average accuracy increased to 84.6%. These values were approximately 10% higher than those obtained when using the training data (Baba et al, 2006; Tong et al, 2020), suggesting that the constructed models have generalized well beyond the training data.

Notably, the model predictions included certain cases that were inconsistent with the experimental training data. For example, both EN and MLP models predicted 22 metabolic reactions as beneficial under glucose condition; however, the experimental training data showed that the removal of each of them did not reduce cell growth on glucose (Tong et al, 2020). The models correctly predicted the essentiality of three genes (*coaA, coaE,* and *hemE*) that were initially thought to be nonessential (Baba et al, 2006), but later found to be essential (Yamamoto et al, 2009). In additional growth experiments, 16 of 19 gene deletions led to growth defects or reduced growth (Fig. 2C). Considering these validated predictions, the accuracy of the EN and MLP models for the glucose condition, increased from initial estimates of 80.7% and 78.7% in model accuracy to 96.6% and 93.7%, respectively. This demonstrates that our models are robust to experimental uncertainty in the training dataset.

Although deletions of nine specific genes were missing from the training growth data, our models predicted that these genes would

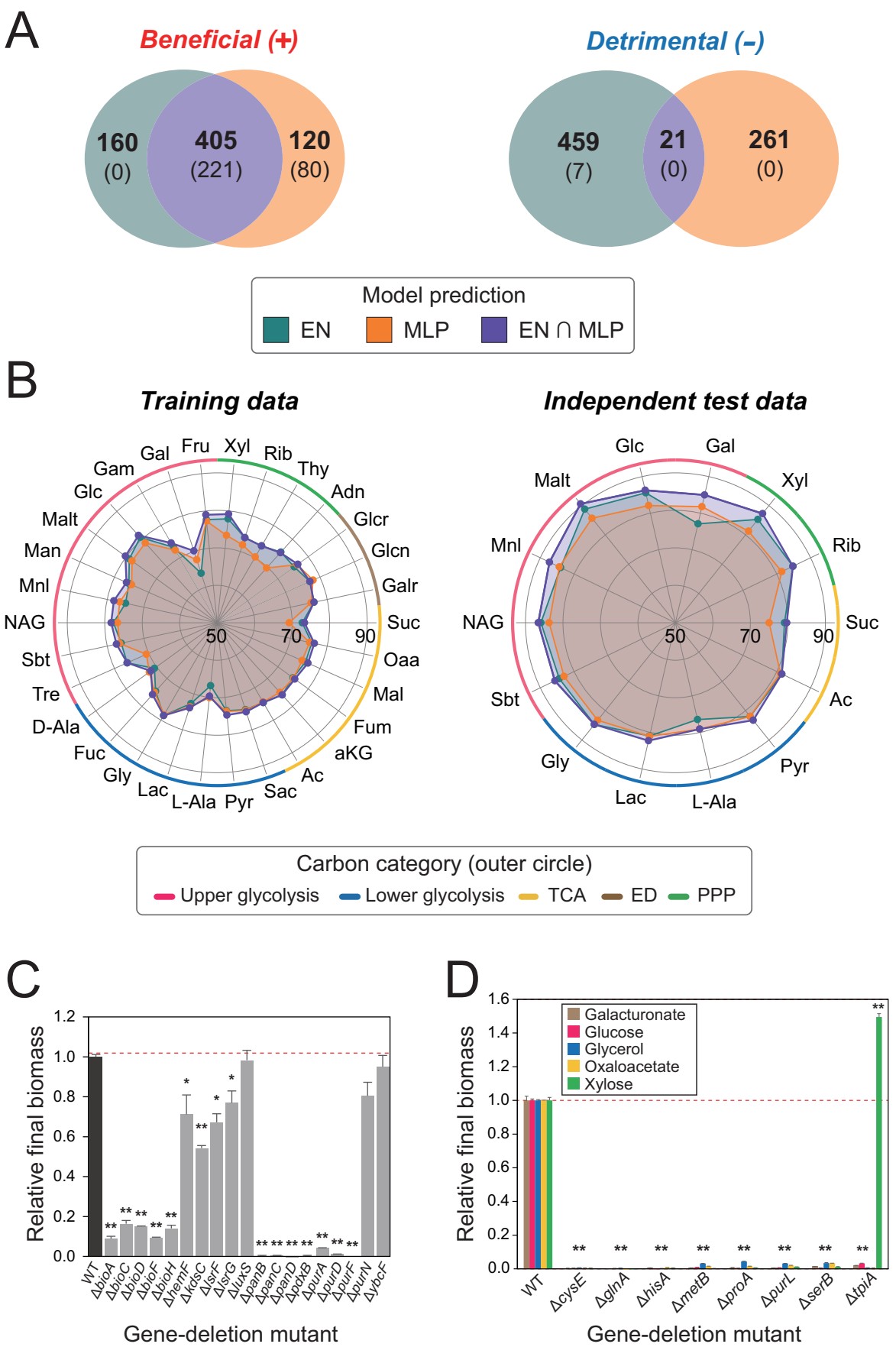

◀ **Figure 2. Evaluation of model accuracy.**

(A) Numbers of metabolic reactions predicted by the elastic net (EN) and multilayer perceptron (MLP) models to be beneficial or detrimental to cell growth under at least one of 30 carbon conditions. The brackets indicate the number of reactions that were predicted under all 30 carbon environments. (B) Assessment of model accuracy (%) by comparing the predicted beneficial reactions with critical reactions reported in the training data (Tong et al, 2020; Baba et al, 2006) (shown on the left), and independent test data (Monk et al, 2017) (right). Carbon sources were categorized according to pathway intermediates, and their abbreviations are explained in Appendix Text S3. (C) Experimental validation of predicted beneficial reactions that are inconsistent with experimental training data for glucose condition. The wild-type (WT) strain and gene-deletion mutants were cultured in MOPS minimal medium supplemented with glucose at 37 °C for 24 h. The y axis denotes the final biomass of a strain relative to that of WT. Asterisks indicate significant differences between the single-gene deletion and WT strain (Student's t test *P value < 0.05, **P value < 0.01). Error bars represent the standard error of the mean (SEM) from three independent cultivations. (D) Experimental validation of predicted beneficial reactions that had not been examined for their deletions in the training data. Cells were cultured in MOPS medium supplemented by different carbon sources (galacturonate, glucose, glycerol, oxaloacetate, or xylose) at 37 °C for 48 h. The y axis, asterisks, and error bars represent the same as in (C).

be beneficial for growth on all 30 carbon sources. These genes have been previously reported to be nonessential (Baba et al, 2006), although one of them, *thyA*, was later found to be essential (Goodall et al, 2018). To validate these predictions, we grew deletions of these genes in minimal medium supplemented with different carbon sources (Fig. 2D). The results showed that deletions of seven genes (*cysE, glnA, hisA, metB, proA, purL,* and *serB*) did not lead to growth on any of the tested carbon sources, including glucose (representing intermediates in upper glycolysis), glycerol (lower glycolysis), galacturonate (Entner-Doudoroff pathway), oxaloacetate (TCA cycle), and xylose (PPP). The deletion of *tpiA* encoding triosephosphate isomerase resulted in growth defects on the tested carbon sources, with xylose being the exception. Although Δ*tpiA* was not included in the training growth data, it was reported to be unable to grow on the agar plate containing minimal xylose medium (Monk et al, 2017). Moreover, considering the training data also used solid agar media, this prediction is hard to be considered a false positive. The notable growth of Δ*tpiA* on xylose, despite an extended lag time in liquid culture (Fig. EV1), warrants further genetic and molecular investigations, as Δ*tpiA* has been reported to be incapable of growing on glucose and glycerol due to the accumulation of highly toxic methylglyoxal (Velur Selvamani et al, 2014).

## Comparison of the model predictions and single-reaction deletion simulations

Single-reaction deletion simulation is a computational approach that is widely used to predict reaction essentiality (Fong and Palsson, 2004; Joyce and Palsson, 2008). Different optimization methods of FBA and MOMA for single-reaction deletion simulations predicted the same essential reactions, which were then compared with the beneficial reactions from the EN and MLP models (Fig. EV2; Dataset EV2). Based on SHAP interpretation, the MLP model predicted 301 metabolic reactions to be beneficial for all 30 carbon conditions. Remarkably, these MLP predictions included all of the 221 reactions predicted by the EN model as well as all of the 214 essential reactions predicted by the single-reaction deletion simulation. The 200 reactions shared among FBA, EN, and MLP included 66 reactions that were previously identified as important or essential in the experimental studies (Baba et al, 2006; Tong et al, 2020). An additional 87 reactions were predicted to be beneficial by either the EN or the MLP models and were not identified by FBA.

Notably, of the 221 beneficial reactions shared by the EN and MLP models, 21 were not predicted by single-reaction deletion

simulation. Among these, the deletion of three reactions (AI2K, PAI2I, and PAI2T) could not be compensated for by other reactions. Among the 18 reactions having alternative pathways, deletions of 12 reactions (4HTHRK, DHORD2, DMPPS, G3PAT160, G3PAT161, IMPD, IPDPS, GMPS, OHPBAT, PAPSR, PERD, and TRPS3) had been experimentally reported to reduce cell growth in minimal glucose media, whereas deletions of six reactions (4HTHRA, TRPAS2, TRPS3, ASNS2, GARFT, and CYTK1) had not (Baba et al, 2006). This result demonstrates that our models can capture the inherent complexity and nonlinearity of biological systems by effectively integrating the experimental data and flux predictions from FBA.

## Functional enrichment analysis of the model predictions

To understand the metabolic features of the model predictions, the metabolic reactions predicted to be beneficial (405 ea) or detrimental (21 ea) to cell growth by both EN and MLP were functionally classified using the Kyoto Encyclopedia of Genes and Genomes Orthology (KO) system (Kanehisa et al, 2023). Of the 405 predicted beneficial reactions, 296 were significantly overrepresented in 6 out of 12 subcategories of KO metabolism ($P \leq 0.01$; Fig. 3A). "amino acid metabolism (A)" had the highest number of predicted reactions (89 ea, 26% of total predicted beneficial reaction), followed by "metabolism of cofactors and vitamins (V)" (86 ea), "carbohydrate metabolism (C)" (82 ea), "nucleotide metabolism (N)" (43 ea), "glycan biosynthesis and metabolism (G)" (26 ea), and "energy metabolism (E)" (24 ea). Fourteen of the 21 detrimental reactions were associated with "carbohydrate metabolism", which was the only significantly overrepresented ($P \leq 0.01$) predicted detrimental reaction without specific enriched metabolic pathways detected. Among the predicted beneficial metabolic pathways, biosynthetic pathways (A, V, N, and G) were highly overrepresented across all 30 carbon sources (Fig. 3B). In contrast, catabolic pathways (C and E) were overrepresented depending on the carbon source. For example, pyruvate and glyoxylate metabolisms were predicted to be beneficial only under L-alanine condition. Classification and distribution analyses provided insights into the relative importance of various metabolic pathways for bacterial cell growth.

Model predictions were further investigated by mapping them onto a major metabolic pathway map (Figs. 4A and EV3A; Appendix Fig. S2). Of the 35 reactions in the central catabolic pathways (glycolysis, tricarboxylic acid cycle [TCA] cycle, and the pentose phosphate pathway [PPP]), 33 were predicted to be beneficial or detrimental, depending on the carbon source.

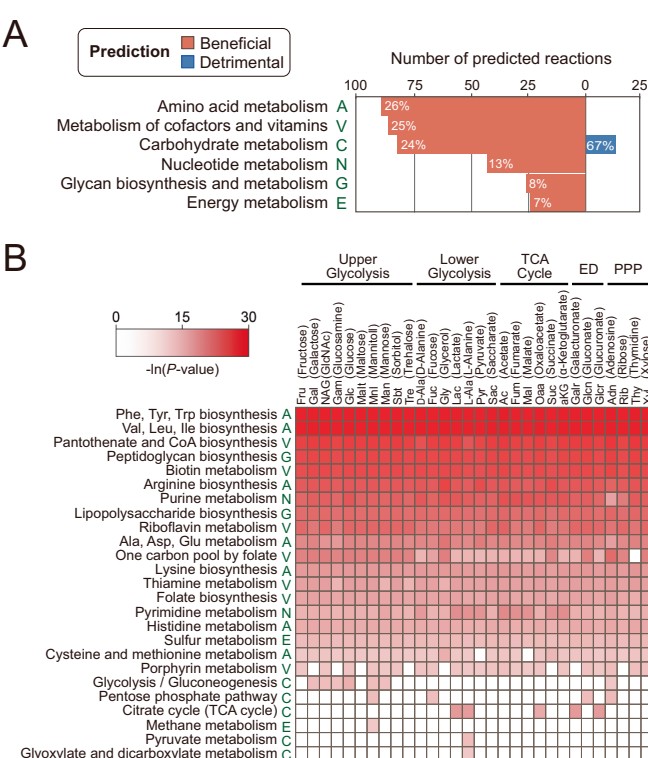

**Figure 3. Functional classification of metabolic reactions predicted to be beneficial or detrimental to cell growth.**

(A) Distribution of metabolic reactions predicted by both EN and MLP models based on Kyoto Encyclopedia of Genes and Genomes (KEGG) orthology classification. The number in each bar denotes the percentage of the predicted reactions. (B) Functional enrichment of the beneficial predictions made by both EN and MLP according to the subcategories of KEGG metabolism and the carbon source. KEGG categories significantly enriched for the model predictions (hypergeometric P value ≤ 0.01) are shown.

However, the majority of metabolic reactions involved in the biosynthetic pathways were predicted to be beneficial for all carbon sources. These include pathways for synthesizing amino acids, purines, pyrimidines, and lipopolysaccharides. Notably, 21 reactions were predicted to be detrimental for at least one carbon source (Figs. 4B and EV3B). These reactions were associated with catabolic pathways (EDA, EDD, G6PDH2r, GCALDD, GLYCTO2, ICL, MALS, PDH, PGI, PGL, PPM, and TALA), gluconeogenesis (PPS and ME1), purine biosynthesis (GART and PRPPS), biosynthesis of vitamins and cofactors (GLYCL and PPPGO), and other metabolic processes (ALDD3y, FDMO, and NHFRBO). These analyses demonstrated that biosynthetic pathways tend to be beneficial regardless of the carbon source, whereas the beneficiality of energy-generating degradative pathways can vary depending on the specific carbon source.

## Experimental validation of metabolic reactions predicted to be detrimental to cell growth

Metabolic reactions that are detrimental to cell growth are of biotechnological importance because inhibiting them can promote cell growth. Both EN and MLP models predicted 21 metabolic

reactions to be detrimental to cell growth, depending on the type of carbon source used (Fig. 4B). Four of these reactions (GART, ICL, MALS, and PPS) were associated with single genes and predicted to be detrimental to multiple carbon conditions. Notably, two reactions (PDH and PRPPS) have been reported to be essential in minimal glucose media (Kim and Copley, 2007). Four of the six predictions (ICL, MALS, PDH, and GART) were validated by follow-up growth experiments using the corresponding gene-deletion mutants in minimal medium supplemented with different carbon sources (Fig. 5A).

The pyruvate dehydrogenase (PDH) complex, encoded by the *pdhR-aceEF-lpdA* operon, plays a pivotal role in the metabolic interconnection between the Embden–Meyerhof–Parnas (EMP) pathway and the TCA cycle by converting pyruvate into acetyl-CoA (Patel and Roche, 1990). The EN and MLP models predicted that PDH is beneficial to cell growth under 14 and 29 carbon conditions, respectively, but detrimental to growth only in acetate. As suggested by these models, ΔaceE did not grow on oxaloacetate or succinate (Fig. 5B). However, it showed improved growth on acetate compared to the wild-type strain, probably because the lack of PDH activity may lead to a switch in metabolism favoring gluconeogenesis to produce the necessary carbon intermediates from acetate, thereby promoting better growth in an acetate-rich environment.

Isocitrate lyase (ICL) and malate synthase (MALS) are involved in the glyoxylate shunt, which is an alternative to the tricarboxylic acid cycle that allows cells to grow on acetate or fatty acids as carbon sources (Dolan and Welch, 2018). The EN and/or MLP models predicted that these reactions would positively affect cell growth under D/L-alanine and acetate conditions, and negatively affect cell growth under fructose, glycerol, and glucose conditions. To block ICL and MALS, we constructed a double-deletion mutant of *E. coli* BW25113 (ΔaceBΔglcB), because ICL and MALS are encoded by aceA and the isozymes aceB and glcB, respectively, and aceB and aceA are expressed from the aceBAK operon. Double-gene deletion resulted in no growth on acetate, and extended the lag time for D-alanine and L-alanine (Fig. 5C). Considering that fatty acids are also degraded directly into acetyl-CoA, which then needs to be metabolized via the glyoxylate shunt (Dolan and Welch, 2018), growth defect of ΔaceBΔglcB on fatty acids can be inferred from the model. As expected, ΔaceBΔglcB did not grow in the minimal medium supplemented with oleic acid as the sole carbon source (Appendix Fig. S3). The extended the lag time of ΔaceBΔglcB growing on D-alanine and L-alanine is intriguing because alanine is purely a glucogenic amino acid which are not degraded directly into acetyl-CoA. Under fructose, glycerol, and glucose conditions, the double-deletion strain showed a growth rate 1.2-fold higher than the growth rate of the parental wild-type strain.

*E. coli* possesses two distinct phosphoribosylglycinamide (GAR) transformylases, GARFT (encoded by *purN*) and GART (*purT*), which convert GAR to 5'-phosphoribosyl-N-formylglycineamide (FGAR) in the purine biosynthesis pathway. These are alternative reactions in that, as a formyl donor, GARFT utilizes $N^{10}$-formyl-tetrahydrofolate, while GART utilizes formate in an ATP-dependent reaction. Notably, GARFT was predicted to be beneficial regardless of the carbon source, whereas GART was predicted to be detrimental for eleven carbon sources. This prediction was tested by growing ΔpurN and ΔpurT in the minimal medium supplemented with each of six carbon sources (gluconate, glucose,

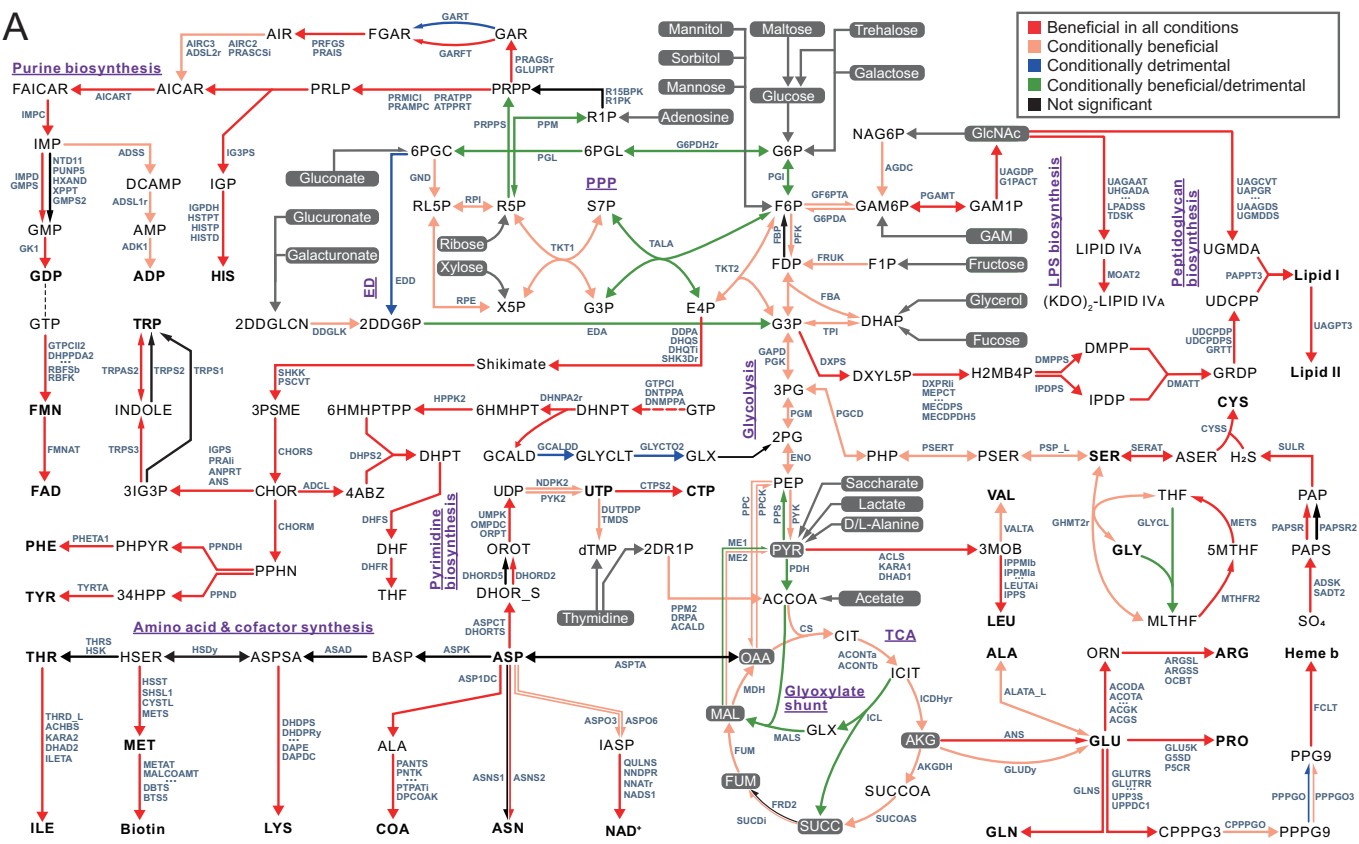

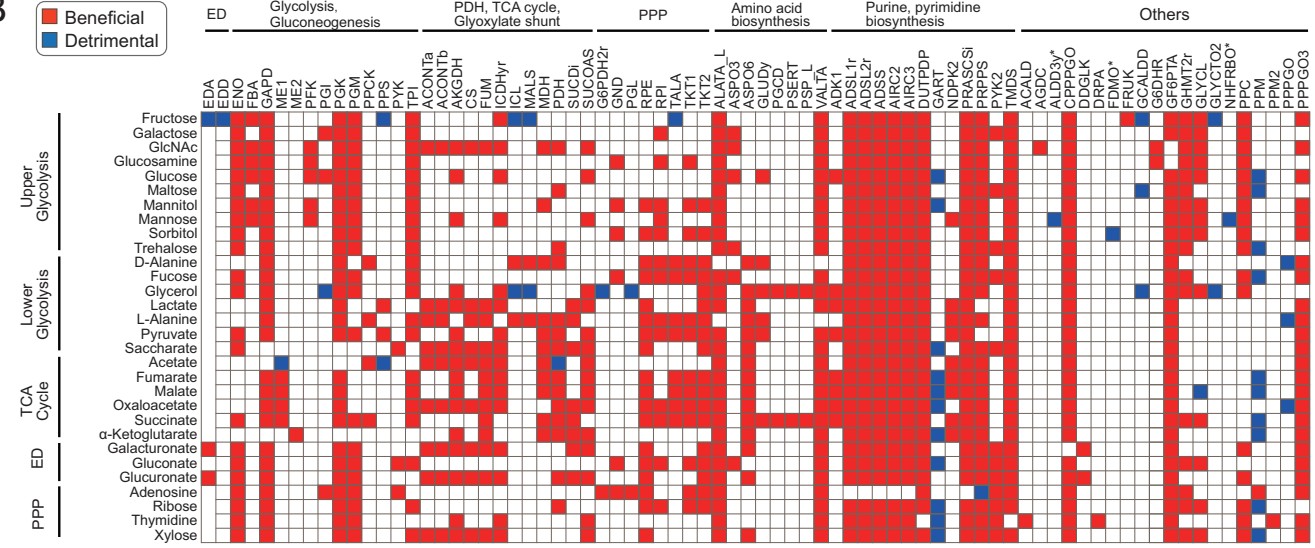

**Figure 4. Effect of the predicted reactions on cell growth.**

(A) Mapping of the model predictions onto metabolic pathways. The solid arrows indicate single metabolic reactions, and the dashed arrows represent multiple sequential reaction steps. Carbon sources are shown with a grey background. The color scheme of each arrow indicates the impact of each metabolic reaction on cell growth: red, beneficial under all 30 carbon source conditions; orange, beneficial under certain carbon conditions; blue, detrimental under certain carbon conditions; green, beneficial or detrimental under certain carbon conditions; black, not significant. (B) Heatmap representation of the impact of the model predictions on cell growth according to the carbon source. Within each cell, the predicted beneficial and detrimental reactions are colored red and blue, respectively. Reactions not shown in (A) are marked with an asterisk. For clarity, reactions that were predicted to be beneficial across all carbon source conditions are not shown.

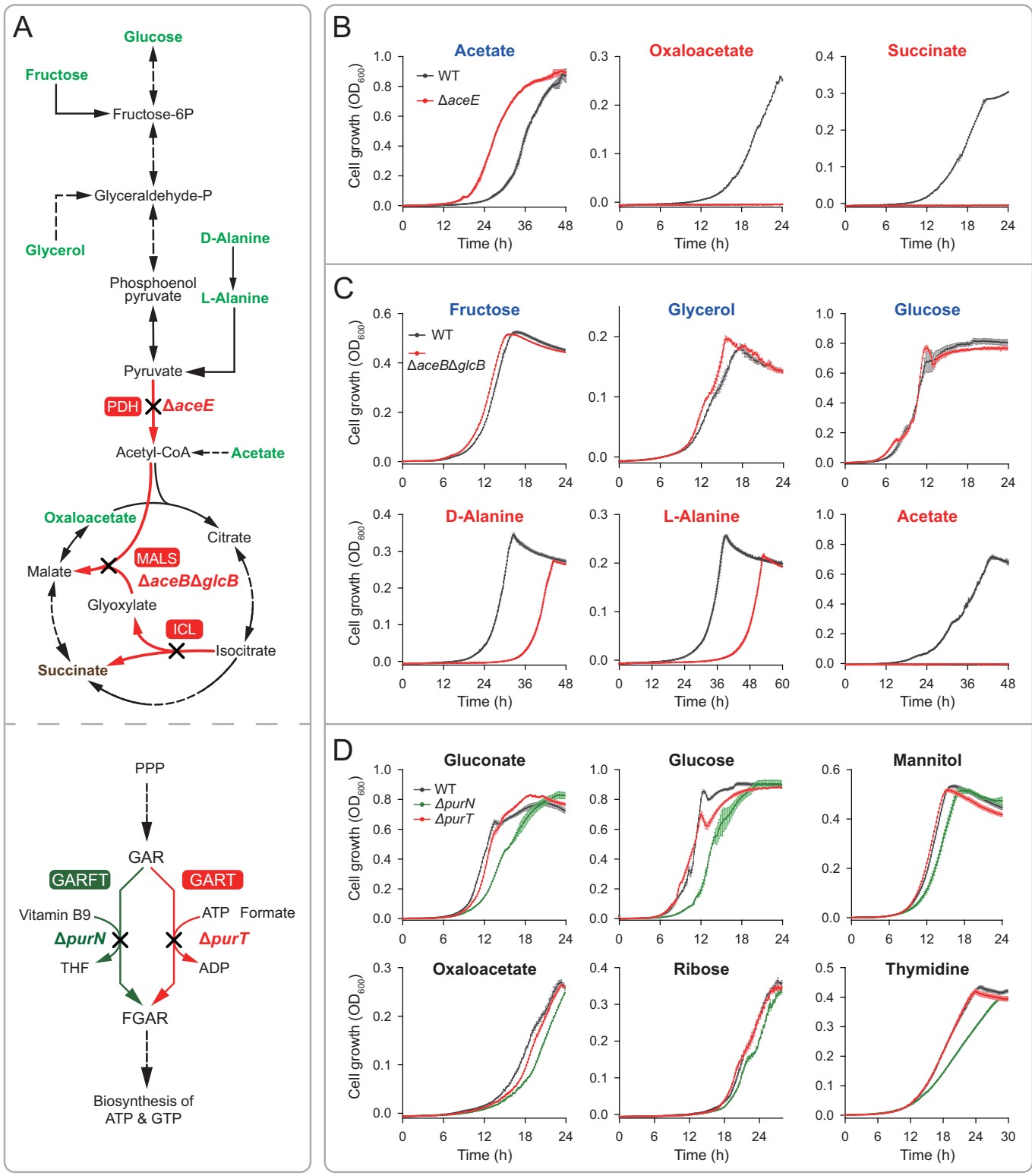

mannitol, oxaloacetate, ribose, and thymidine). In all carbon sources tested, Δ*purT* grew equally well as the wild-type strain, while Δ*purN* showed retarded cell growth (Fig. 5D). We did not test the double-deletion strain of *purN* and *purT* as it requires an exogenous purine source for growth (Nygaard and Smith, 1993).

These results indicate that the loss of either *purN* or *purT* can be compensated for by the remaining gene and *purN* is more central to de novo purine biosynthesis. However, these validated predictions are not immediately explicable and require more in-depth biochemical and genetic investigations.

**Figure 5. Experimental validation of metabolic reactions predicted to be detrimental to cell growth depending on the carbon source.**

(A) Reaction steps subjected to growth curve experiments of gene-deletion mutants. Metabolic reactions blocked by the gene deletion (marked by an X) are shown in a colored background. The solid arrows indicate single metabolic reactions, and dashed arrows represent multiple sequential reaction steps. The upper part corresponds to (B, C), and the tested carbon sources are colored green. The lower part corresponds to (D). (B–D) Growth curves are shown for mutant strains deficient of pyruvate dehydrogenase (PDH) (B), isocitrate lyase (ICL) and malate synthase (MALS) (C), and two phosphoribosylglycinamide (GAR) transformylases (GARFT and GART) (D). As a control, growth curves for the wild-type (WT) parental strain (*E. coli* K-12 BW25113) are shown. Carbon source conditions where the reaction (colored red in (A)) was predicted to be beneficial or detrimental to cell growth are colored red or blue, respectively.

## Discussion

In this study, ML models were developed to pinpoint metabolic reactions that markedly influence bacterial growth under different carbon source conditions. The basic rationale for the causal predictions is that the GEM underpinning flux simulations is mechanistically constructed; thus, the features constituting the trained ML models become mechanistically causal (Sahu et al, 2021; Yang et al, 2019). ML models learn patterns and relationships between input data and output, and whether the predictions are causal or correlative depends on the nature of the input data and the underlying relationships. If the input features are primarily observational (such as omics data), ML models typically identify correlations or associations between input features and the output variable. To make causal predictions (meaning that changing input features will directly cause changes in the output), input features need to represent causative factors for the output variable. A GEM is a mathematical representation of interconnected metabolic reactions and compounds, facilitating metabolic simulations of how an organism transforms nutrients into energy and other necessary molecules. Because the GEM-based fluxes as the input features are derived from the mechanistically constructed GEM, the ML models are expected to make causal predictions that reflect the direct influence of metabolic reactions on growth outcomes in different carbon conditions. The qualitative input–output relationships were learned and quantified by assigning specific numerical values, and we determined which input feature reactions had statistically significant relationships with the output data.

The choice of the modeling method depends on the characteristics of the data, the interpretability of the results, and modeling capability (Sarker, 2021). In this study, two distinct supervised learning methods, EN and MLP, were employed to capture different aspects of the data and ensure robust analysis. The EN and MLP methods represent different ends of the interpretability-flexibility spectrum in ML. EN, a form of linear regression (Zou and Hastie, 2005), is effective at dealing with multicollinearity, which was the correlation between independent variables (metabolic fluxes in this study). This property is crucial for analyzing metabolic networks, where multiple reactions are often interlinked and their fluxes can be highly correlated. The coefficients provided by the EN model offer a straightforward interpretation of the relative importance of each metabolic reaction in cell growth, making it a "white-box" model. MLP, a type of artificial neural network (Gardner and Dorling, 1998), is well-suited for capturing complex, nonlinear relationships between input data (metabolic fluxes) and output data (cell growth). The MLP model can detect more complex patterns and interactions in data that may be missed by a linear model, such as EN. However, owing to its complex architecture, the MLP is often referred to as a "black-box" model, where it is challenging to understand how inputs relate to outputs.

To enhance the interpretability of the MLP model, we employed the SHAP interpretation method (Lundberg and Lee, 2017), which assigns importance values to the input features and estimates the contribution of each metabolic reaction to cell growth. Combining these two models provides a reliable and interpretable foundation for understanding the complex interplay between metabolic reactions during cell growth.

This study leveraged the power of two ML methods, EN and MLP, to predict the impact of metabolic reactions on bacterial growth. We combined the phenotypic data from a genome-wide gene-deletion library with simulated flux distributions to create a comprehensive dataset that captured the complexity of *E. coli* metabolism across a wide range of carbon sources (Fig. 1). This integrated dataset served as a foundation for training both the EN and MLP models. These models aim to learn the intricate relationships between the input data (flux distributions) and output data (growth phenotypes). Importantly, the flux predictions were grounded in a genome-scale metabolic model, lending a mechanistic and causal layer to the predictive models, which set them apart from traditional statistical models. Another significant advantage of our ML models is their ability to identify not just essential metabolic reactions, but also nonessential ones that promote growth (Fig. 2C). This extends beyond the scope of traditional reaction deletion simulation, which primarily focuses on predicting essential reactions for cell growth (Joyce and Palsson, 2008). The inclusion of nonessential growth-promoting reactions makes these models potentially more useful for predicting the effects of metabolic manipulations, such as those used in metabolic engineering.

Both EN and MLP models were effective at predicting metabolic reactions that are beneficial to cell growth (Fig. 2A). This is not surprising, considering that bacterial cellular metabolism has evolved to support cell growth; therefore, metabolic reactions are primarily beneficial for cell growth (Ibarra et al, 2002; Nielsen, 2007). Interestingly, a significant proportion of the beneficial predictions (97% on average per carbon) from the MLP model agreed with those from the EN model (Appendix Table S2). This high convergence indicates that both modeling techniques effectively capture similar patterns in the data, such as multicollinearity. This also indicates that both models appropriately modeled the underlying structure of the metabolic network, providing confidence in their predictive capabilities. However, the low number of detrimental predictions from both models may indicate data bias toward beneficial reactions. This aspect is worth exploring further, perhaps by introducing different types of interventions such as the overexpression of each gene.

Although the EN and MLP models were trained using the same dataset and feature selection method, they showed high discrepancy in predicting detrimental reactions (Fig. 2A). This was especially evident when the EN model alone identified seven reactions as

detrimental across all carbon source conditions. This discrepancy can be attributed to the characteristics of the output data and the different methods employed by the EN and MLP models to calculate the feature importance. The training output dataset examining cell growth of gene-deletion mutants (Baba et al, 2006; Tong et al, 2020), displayed much more occurrences of reduced biomasses than increased ones. Moreover, the extent of biomass reduction, sometimes even resulting in no growth, was not only more pronounced but also displayed a wider variability than the cases of growth enhancement. This data bias towards reduced cell growth can lead the MLP model to assign high positive SHAP values to more reactions. As SHAP values are calculated within the context of feature interactions and dependencies (Lundberg and Lee, 2017), features with such highly positive SHAP values might dominate, potentially overshadowing or constraining the influence of features with negative SHAP values. On the other hand, the regression coefficients in the EN model are not heavily influenced by feature interactions, which might make it better suited to predict detrimental reactions compared to the MLP model.

The ML models predicted that the majority of reactions involved in biosynthetic pathways, especially in amino acid synthetic pathways, are beneficial to cell growth regardless of the carbon source, while involved in catabolic pathways, especially in central carbon metabolism, became beneficial or detrimental, depending on the carbon source utilized (Figs. 3 and 4). This discrepancy can be attributed to how these pathways are structured and how the carbon source is mainly catabolized through the metabolic network. Catabolic pathways typically are highly interconnected, offering multiple alternative routes for carbon source utilization (Kim and Copley, 2007). For example, the upper EMP pathway was predicted to be beneficial for carbon sources such as glucose, N-acetyl glucosamine, fructose that enter through that pathway. However, for carbon sources like acetate, α-ketoglutarate, which follow different catabolic routes, EMP pathway was not predicted to be beneficial. In contrast, biosynthetic pathways tend to be linear and require activation in the absence of externally supplied amino acids or other essential building blocks. This interpretation of the model predictions underlines the importance of the structural character-istics of metabolic pathways in understanding how microorganisms respond to nutritional changes, requiring further in-depth inves-tigation to unravel the exact underlying mechanisms.

The consistency of the model predictions with the results from follow-up experiments demonstrates the power and potential of our models in unraveling the complex relationships between metabolic reactions and cell growth. Validation of the model predictions against experimental data proved the robustness of the models (Fig. 2C). This is crucial, as experimental data can contain noise due to various factors, such as measurement error, biological variability, and uncontrolled conditions. Our models predicted the essentiality of genes whose deletions were not tested in the output data of the training dataset (Fig. 2D). Despite being beyond the scope of the experimental output data, the feature fluxes associated with these genes are still included in the simulated input data, providing necessary information for the models to make predic-tions about those genes' essentiality. Thus, these cases should fit the concept of interpolation rather than extrapolation. The experi-mental validation of the predictions regarding the effects of different carbon sources on metabolic reactions demonstrated the potential of the model for scientific discoveries (Fig. 5). In particular, the impact of various carbon sources on metabolic reactions, such as glyoxylate shunt (ICL and MALS), the link between the EMP pathway and the TCA cycle (PDH), and a redundant reaction in the purine biosynthesis pathway (GART), imply intricate regulatory mechanisms and interactions. Although these interactions are not yet fully understood, their identification indicates the layers of metabolic regulation that adapt to environmental changes, including different carbon sources.

Bacteria adjust their metabolic processes based on available carbon sources to ensure their survival and proliferation (Litsios et al, 2018; Shimizu, 2016). The predictive models developed in this study provide insights into which metabolic reactions influence growth in the presence of different carbon sources. Although their ability to explain the underlying metabolic and regulatory mechanisms is limited, the intriguing predictions, including the detrimental reactions in central catabolic pathways, serve as hypotheses that guide further experimental investigations, enhan-cing our knowledge of cellular metabolism and its role in cell growth. The explainable nature of these models is valuable for new scientific discoveries and practical applications, such as the optimization of growth conditions and the design of microbial cell factories for the development of various bioprocesses. Moreover, the approach employed in this study can be readily applied to other organisms and adapted to study different cellular traits.

## Methods

### Bacterial strains and culture conditions

The *E. coli* strains used in this study are listed in Appendix Table S3. Single-gene-deletion mutants of *E. coli* K-12 BW25113 were obtained from the Keio collection (Baba et al, 2006). Cells were cultured as previously described (Tong et al, 2020). Seed cultures were prepared by growing cells overnight in Luria-Bertani (LB) medium (10 g/L tryptone, 5 g/L yeast extract, and 10 g/L NaCl) and subculturing in LB medium until the early exponential phase. After washing with MOPS minimal medium (Teknova, Hollister, CA) without a carbon source, the cell cultures were diluted 1:100 in MOPS medium supplemented with the carbon source of interest. The main cultures were performed in a 96-well Epoch 2 Microplate Spectrophotometer (BioTek, Winooski, VT) containing 150 µL of culture medium in each well. The concentra-tions of each carbon source were 60 mM acetate, 40 mM D-alanine, 20 mM fructose, 20 mM galacturonate, 20 mM gluconate, 20 mM glucose, 40 mM glycerol, 40 mM L-alanine, 20 mM mannitol, 30 mM oxaloacetate, 24 mM ribose, 30 mM succinate, 12 mM thymidine, and 24 mM xylose, as described previously (Tong et al, 2020). All cell cultures were performed at 37 °C and the optical density at 600 nm ($OD_{600}$) was measured every 10 min. The specific growth rate and lag time for growth curve experiments were calculated using AMiGA software (Midani et al, 2021).

### Construction of a double-deletion mutant

The *aceB* gene was deleted from the single-gene-deletion mutant (*E. coli* BW25113Δ*glcB*) obtained from Keio collection using the λ Red recombinase system method (Datsenko and Wanner, 2000). Briefly, the chloramphenicol resistance (Cmr) cassette was

amplified from the plasmid pKD3 (ATCC, Manassas, VA) using primers for *aceB* deletion (forward primer, 5' TCGTTCACAGTGGGGAAGTTTTCGGATCCATGACGAGGAGCTGCACGATGGTGTAGGCTGGAGCTGCTTC 3'; reverse primer, 5' GTGCAGATGCTCCATAGTTATGTGGTGGTTTACGCTAACAGGCGGTAGCCATGGGAATTAGCCATGGTCC 3'). PCR products were electroporated into an electrocompetent Δ*glcB* mutant harboring pKD46 (ATCC). After induction with L-arabinose (1 mM) at an $OD_{600}$ of 0.3, colonies were selected on LB agar plates containing chloramphenicol (25 µg/mL). Gene replacement with the *cat* gene was confirmed by PCR (forward primer, 5' CTGGTGACGCATTTTACGCC 3'; reverse primer, 5' TTTCACATTGGCGTTGAGCG 3').

## Genome-scale metabolic modeling and simulation

iML1515, the GEM of the *E. coli* K-12 MG1655 (Monk et al, 2017), was used for metabolic simulations. To allow the GEM to simulate oxaloacetate utilization, we added oxaloacetate (oaa) and three metabolic reactions (BiGG ID: EX_oaa_e, OAAt2_2pp, and OAAtex) contained in *E. coli* K-12 DH10B GEM (iECDH10B_1368) (Monk et al, 2013) to iML1515. Several genes (*araBAD*, *rhaBAD*, and *lacZ*) present in iML1515 are absent in *E. coli* K-12 BW25113 (Monk et al, 2017); therefore, the fluxes of their associated reactions (ARAI, RBK_L1, RMPA, LYXI, RMI, RMK, and LACZ) were constrained to zero. The modified iML1515 comprised 1516 genes, 1879 unique metabolites, and 2715 metabolic reactions. Metabolic simulations were performed using the Python package COBRApy (Ebrahim et al, 2013) and Gurobi Optimizer version 9.1.2 (Gurobi Optimization, Beaverton, OR). To simulate growth using in silico MOPS minimal medium, the uptake of each metabolite present in MOPS medium was allowed by unconstraining the reaction bounds of the corresponding exchange reactions (Appendix Table S4). The maximum uptake rates of the carbon source and oxygen were set at 10 and 20 mmol/gDCW/h, respectively (Monk et al, 2017; Tong et al, 2020). To prepare the input data for model construction, MOMA (Segrè et al, 2002) was used to simulate metabolic flux distributions for each gene deletion, which were performed in triplicate, and the flux values were averaged. Single-reaction deletion simulations were performed using FBA and MOMA, and a reaction was considered essential if removal from the model reduced the growth rate to < 5% of the growth objective value calculated for the wild-type parental strain.

## Construction of an elastic net regression model

Using the elastic_net function in the H2O4GPU software (Gill et al, 2021), EN regression was performed with the biomass and simulated metabolic fluxes of individual gene-deletion mutants. The parameters in the model formulation were selected to determine the minimum root mean square error of the fitted model (Appendix Table S5). The parameters chosen for the regression were an L1-ratio of 0.01, 300-fold cross-validation, a maximum of 1e4 iterations, and a tolerance of 1e-6. After running the regression, the metabolic reactions were divided into two groups based on the sign (positive or negative) of their coefficients. Reactions with coefficients greater than one-tenth of the standard deviation of each group were selected. Exchange and transport reactions were excluded from the analyses.

## Construction of a multilayer perceptron model

The MLP model, a fully connected feedforward neural network, was constructed using the Keras (Chollet, 2017) and TensorFlow (Abadi et al, 2016) packages. The input layer consisted of the simulated metabolic fluxes of the individual gene-deletion mutants. The output layer was a single node representing the biomasses of the gene-deletion mutants, which were connected to the previous layer through a rectified linear unit (ReLU) activation function. Optimal model hyperparameters were determined using the RandomSearch function in the KerasTuner Python package (O'Malley et al, 2019) (Appendix Table S5). The tuning process was performed under glucose conditions with a maximum of 10,000 trials and a validation set size of 10%. A fixed random seed was used to ensure the reproducibility of the results. The optimized hyperparameters were four hidden layers, each with 1000 nodes linked through a ReLU activation function; a dropout rate of 0.6 (used to prevent overfitting); a kernel constraint of 4 (to regularize the weights); and an RMSprop optimizer (to minimize the loss function).

The DeepExplainer function in SHAP Python package (Lundberg and Lee, 2017) was applied to calculate the importance of each metabolic reaction (feature) in the MLP model using the entire training dataset as the background sample. The MLP model was trained and interpreted with SHAP ten times with different random seeds. The median SHAP value for each feature in each run was averaged to determine the importance of each metabolic reaction. Reactions were classified based on the sign (positive or negative) of the SHAP values. Reactions with coefficients greater than one-tenth of the standard deviation of each group were selected. Exchange and transport reactions were excluded from the analyses.

## Measurement of model accuracy

The accuracy of EN and MLP models were separately measured against two datasets of the training dataset (Baba et al, 2006; Tong et al, 2020) and an independent dataset (Monk et al, 2017). In the studies reporting the datasets (Monk et al, 2017; Tong et al, 2020), the authors defined critical genes by their own criteria. To retrieve critical reactions in each dataset, those reported critical genes (Dataset EV2) were mapped to the corresponding metabolic reactions using the gene-protein-reaction (GPR) rules included in the iML1515 model (Monk et al, 2017). As the critical reactions were determined based on the phenotypic screen of the single-gene deletions, each reaction is highly likely to be directly influenced by the absence of a specific gene. Thus, for a fair comparison, among the beneficial reactions predicted by each model, we only considered metabolic reactions that are mediated by a single protein or protein complex and are not otherwise activated by other proteins. The prediction accuracy of each model against a specific dataset was calculated as follows:

$$Prediction\ accuracy = P(B/A) = \frac{number\ of\ (A \cap B)}{number\ of\ A},$$

where *A* is the beneficial reactions predicted by a model, and *B* is the critical reactions reported in an experimental dataset.

## Functional enrichment analysis

Functional enrichment analysis was performed to identify over-represented metabolic pathways within the model predictions using the metabolism category of the KO system (Kanehisa et al, 2023). To evaluate the enrichment of each pathway, the hypergeometric *P* value and cumulative probability were calculated using the hypergeometric function in the SciPy Python package (Virtanen et al, 2020). Metabolic pathways with a *P* value ≤ 0.01 and a cumulative probability ≥0.95 were considered significantly overrepresented.

## Data availability

All data, models, and codes used in this study are available on GitHub (https://github.com/sybirg/xai_growth) with a DOI of Zenodo (https://zenodo.org/records/10164986), along with information for replicating the presented results.

## Peer review information

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

## Acknowledgements

This work was supported by the National Research Foundation of Korea (NRF) grant funded by the Korea government (MSIT) [NRF-2022R1A2C1003800 and NRF-2018R1D1A1B07044848].

## Author contributions

**Hyunjae Woo**: Data curation; Software; Formal analysis; Validation; Investigation; Visualization; Methodology; Writing—original draft; Writing—review and editing. **Youngshin Kim**: Validation; Investigation; Methodology; Writing—review and editing. **Dohyeon Kim**: Software; Investigation; Methodology; Writing—review and editing. **Sung Ho Yoon**: Conceptualization; Resources; Data curation; Supervision; Funding acquisition; Validation; Investigation; Writing—original draft; Project administration; Writing—review and editing.

## Disclosure and competing interests statement

The authors declare no competing interests.

# Expanded View Figures

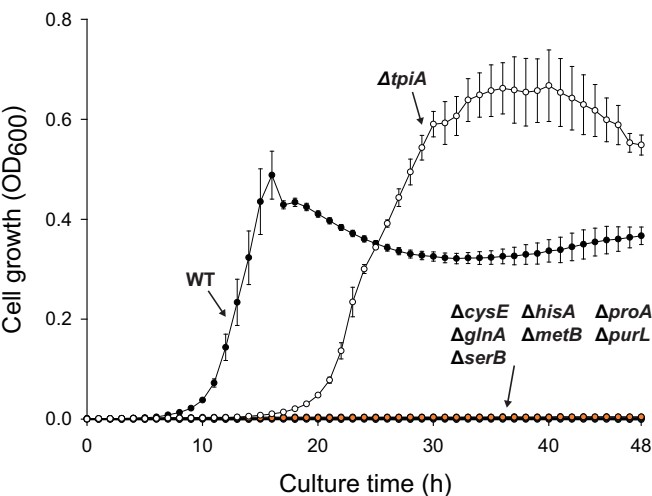

**Figure EV1.  Experimental validation of predicted beneficial reactions with missing deletions in the training data (Xylose condition).**

The *y* axis denotes relative biomasses of gene-deletion mutants grown in the MOPS minimal medium supplemented with xylose as the sole carbon source for 48 h. Growth measurements are represented as the mean ± SEM from three independent cultivations.

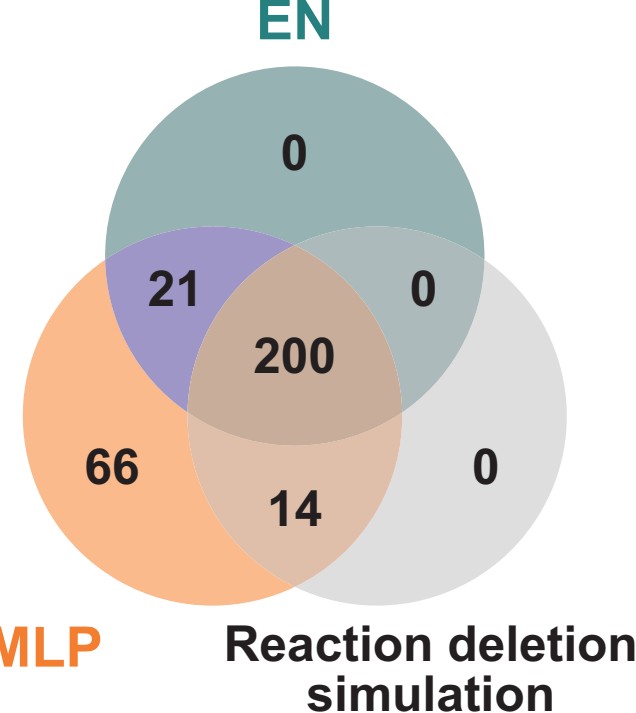

**Figure EV2. Comparison of the predictions from the EN model, MLP model, and the single-reaction deletion simulation.**

Metabolic reactions that were predicted to be beneficial for all 30 carbon conditions by the EN and MLP models were compared with essential reactions predicted from the simulation.

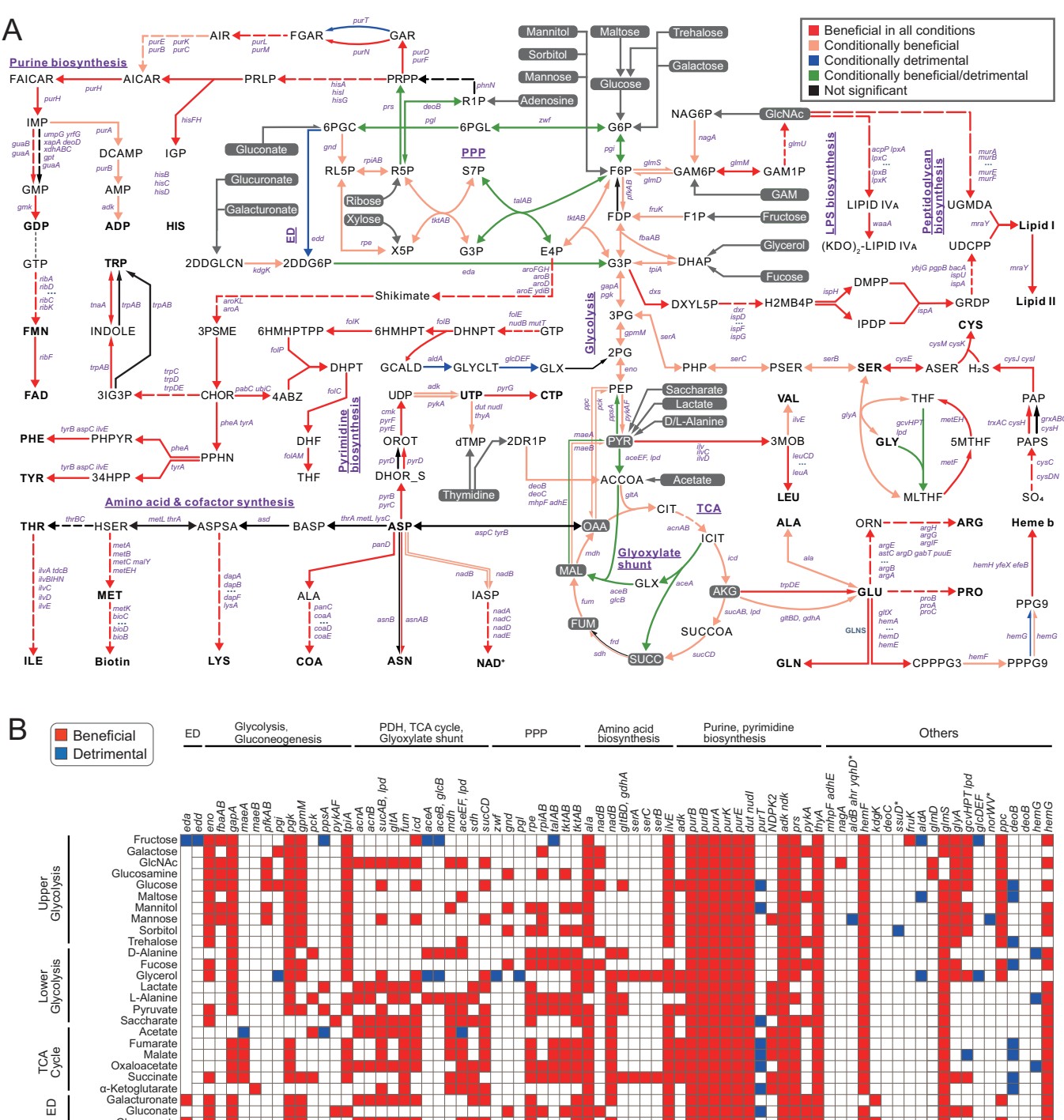

**Figure EV3. Effect of the predicted metabolic genes on cell growth.**

This figure is identical to Fig. 4 except that the gene names are shown instead of the reaction names. (A) Mapping of the model predictions onto metabolic pathways. The solid arrows indicate single metabolic reactions, and dashed arrows indicate multiple sequential reaction steps. Carbon sources are shown with a grey background. The color scheme of each arrow indicates the impact of each metabolic reaction on cell growth: red, beneficial under all 30 carbon source conditions; orange, beneficial under certain carbon conditions; blue, detrimental under certain carbon conditions; green, beneficial or detrimental under certain carbon conditions; black, not significant. (B) Heatmap representation of the impact of the model predictions on cell growth according to the carbon source. Within each cell, the predicted beneficial and detrimental reactions are colored red and blue, respectively. Metabolic genes not shown in panel A are marked with an asterisk. For clarity, genes that were predicted to be beneficial across all carbon source conditions are not shown.

