## [Peer Review File · Molecular Systems Biology]

Machine learning identifies key metabolic reactions in bacterial growth on different carbon sources

Hyunjae Woo, Youngshin Kim, Dohyeon Kim, and Sung Ho Yoon

Corresponding author(s): Sung Ho Yoon (syoon@konkuk.ac.kr)

Review Timeline:

Submission Date:	16th Aug 23
Editorial Decision:	21st Sep 23
Revision Received:	28th Nov 23
Editorial Decision:	14th Dec 23
Revision Received:	3rd Jan 24
Accepted:	11th Jan 24

Editor: Poonam Bheda

Transaction Report:

21st Sep 2023

Dear Dr Yoon,

Thank you again for submitting your work to Molecular Systems Biology. We have now heard back from the three reviewers who agreed to evaluate your study. As you will see below, the reviewers appreciate that the presented approach addresses a relevant problem. However, they raise a series of concerns, which we would ask you to address in a major revision.

I think that the recommendations of the reviewers are rather clear and I therefore do not see the need to repeat the comments listed below. Ensuring that the code is provided in such a way that it can be checked in full is important for the review process. All other issues raised would need to be satisfactorily addressed. Please let me know in case you would like to discuss in further detail any of the issues raised, I would be happy to schedule a call.

We require:

4) A .docx formatted letter INCLUDING the reviewers' reports and your detailed point-by-point responses to their comments. As part of the EMBO Press transparent editorial process, the point-by-point response is part of the Review Process File (RPF), which will be published alongside your paper.

5) A complete author checklist, which you can download from our author guidelines (<https://www.embopress.org/page/journal/17574684/authorguide#submissionofrevisions>). Please insert information in the checklist that is also reflected in the manuscript. The completed author checklist will also be part of the RPF.

6) Please note that all corresponding authors are required to supply an ORCID ID for their name upon submission of a revised manuscript.

7) It is mandatory to include a 'Data Availability' section after the Materials and Methods. Before submitting your revision, primary datasets produced in this study need to be deposited in an appropriate public database, and the accession numbers and database listed under 'Data Availability'. Please remember to provide a reviewer password if the datasets are not yet public (see <https://www.embopress.org/page/journal/17574684/authorguide#dataavailability>).

This study includes no data deposited in external repositories.

8) For data quantification: please specify the name of the statistical test used to generate error bars and P values, the number (n) of independent experiments (specify technical or biological replicates) underlying each data point and the test used to calculate p-values in each figure legend. The figure legends should contain a basic description of n, P and the test applied. Graphs must include a description of the bars and the error bars (s.d., s.e.m.). Please provide exact p values.

10) We replaced Supplementary Information with Expanded View (EV) Figures and Tables that are collapsible/expandable

online. A maximum of 5 EV Figures can be typeset. EV Figures should be cited as 'Figure EV1, Figure EV2' etc... in the text and their respective legends should be included in the main text after the legends of regular figures.

<https://www.embopress.org/page/journal/17574684/authorguide#expandedview>

11) For more information: There is space at the end of each article to list relevant web links for further consultation by our readers. Could you identify some relevant ones and provide such information as well? Some examples are patient associations, relevant databases, OMIM/proteins/genes links, author's websites, etc...

12) Author contributions: CRediT has replaced the traditional author contributions section because it offers a systematic machine readable author contributions format that allows for more effective research assessment. Please remove the Authors Contributions from the manuscript and use the free text boxes beneath each contributing author's name in our system to add specific details on the author's contribution. More information is available in our guide to authors.

13) Disclosure statement and competing interests: We updated our journal's competing interests policy in January 2022 and request authors to consider both actual and perceived competing interests. Please review the policy <https://www.embopress.org/competing-interests> and update your competing interests if necessary.

14) Every published paper now includes a 'Synopsis' to further enhance discoverability. Synopses are displayed on the journal webpage and are freely accessible to all readers. They include a short stand first (maximum of 300 characters, including space) as well as 2-5 one-sentences bullet points that summarizes the paper. Please write the bullet points to summarize the key NEW findings. They should be designed to be complementary to the abstract - i.e. not repeat the same text. We encourage inclusion of key acronyms and quantitative information (maximum of 30 words / bullet point). Please use the passive voice. Please attach these in a separate file or send them by email, we will incorporate them accordingly.

Please also suggest a striking image or visual abstract to illustrate your article as a PNG file 550 px wide x 300-600 px high. Share synopsis text and image, as well as eTOC:

Please note that these would be the final versions and changes during proofing are usually not allowed

15) As part of the EMBO Publications transparent editorial process initiative (see our Editorial at <http://embomolmed.embopress.org/content/2/9/329>), Molecular Systems Biology Medicine will publish online a Review Process File (RPF) to accompany accepted manuscripts.

In the event of acceptance, this file will be published in conjunction with your paper and will include the anonymous referee reports, your point-by-point response and all pertinent correspondence relating to the manuscript. Let us know whether you agree with the publication of the RPF and as here, if you want to remove or not any figures from it prior to publication. Please note that the Authors checklist will be published at the end of the RPF.

Molecular Systems Biology has a "scooping protection" policy, whereby similar findings that are published by others during review or revision are not a criterion for rejection. Should you decide to submit a revised version, I do ask that you get in touch after three months if you have not completed it, to update us on the status.

I look forward to receiving your revised manuscript.

Yours sincerely,

Poonam Bheda, PhD
Scientific Editor
Molecular Systems Biology

Reviewer #1:

Hyunjae et al. propose machine learning models that integrate genome-wide gene-deletion data and simulated flux distributions

to identify metabolic reactions beneficial or detrimental to *Escherichia coli* grown on 30 different carbon sources. The authors use FBA to generate fluxes. They interpret the models using SHAP and experimentally validate the metabolic reactions.

Although the performance is reasonable even if fairly baseline machine learning models have been used, the way in which the features have been generated is not satisfactory, and this is likely to have strong effects on the accuracy that can be achieved by all models.

"Simulate oxaloacetate utilization, three metabolic reactions (BiGG ID: EX_oaa_e, OAAi2_2pp, and OAAtex) were added to the model." Can the authors justify these manual curation steps, maybe adding key references showing that these three reactions are required for oxaloacetate?

"The maximum uptake rates of the carbon source and oxygen were set at 10 and 20 mmol/gDCW/h, respectively" The authors have not clarified how and why they constrain the carbon and oxygen uptake rates to 10 and 20 mmol/gDW/h. Are these values from the experimental data or from a previous study?

Furthermore, the models have not been constrained using relevant media composition or transcriptomics data. Constraining the model is likely to help generate much more accurate features.

The authors simulated the metabolic fluxes using FBA and MOMA, which were used as input features for machine learning and deep learning models. However, the flux rates generated by FBA for most of the reactions are zeros, which are not considered good features for machine learning models. Therefore, it is recommended to use FVA or flux sampling techniques (<https://www.ncbi.nlm.nih.gov/pmc/articles/PMC6718391/>) to generate more accurate features for machine learning models.

"Thus, we assessed model accuracy by comparing the reactions (features) selected by the models with the output data of the training dataset [4, 23] ...". The authors have designed regression models however, they have not clearly mentioned how the accuracy of the model was evaluated. The authors need to provide more details (mathematically would be best) on how exactly the models' accuracy was evaluated.

There seems to be an extremely high discordance between the EN and the MLP approach when predicting the detrimental steps. Given that 7 reactions are predicted under all carbon sources, is it because the EN method is too permissive when selecting key features, or is the MLP too strict?

"Regarding the critical reactions reported in the training data [4, 23], the EN model showed an average accuracy of 74.9% per carbon source,...". A machine learning model with an average accuracy of 74.9% is not sufficient to convince that the proposed framework (CBM with machine learning) is the best to identify the metabolic reactions that significantly influence bacterial cell growth.

Given the final results on the accuracy, a thorough examination of the main contribution presented by the paper is missing. For instance, it has not been investigated whether features could have been generated in a different way. The authors could have tried generating more accurate features (for instance, via FVA or flux sampling techniques), then applying feature selection techniques (e.g., variance threshold, tree-based feature selection) to reduce high dimensional data and remove noise (irrelevant features). Most importantly, the authors need to clarify how the accuracy of the regression models was evaluated, as this would elucidate likely steps that could improve it.

"Supplementary Table S3" is missing.

In the deep learning pipeline, the authors tuned the hyperparameters first and used the same hyperparameters while training the model for ten seeds (iterations). However, in this case, this is not the best way to tune the hyperparameters. It is recommended to tune the hyperparameters for each iteration. While training the model with a smaller sample size, it is best recommended to train the models using nested k-fold validation where the inner fold is used to tune the hyperparameter and train the model with the best hyperparameter and the outer fold is used to evaluate the model (<https://www.mdpi.com/2076-3417/12/13/6681>).

The code could not be checked in full. The "deep_learning.ipynb" script throws an exception at line: `X_train_scaled, y_train = sklearn.utils.shuffle(X_train_scaled, y_train, random_state=random_seed)`. It seems that the author missed defining the `X_train_scaled` and `y_train` variables.

"The MLP model predicted 301 metabolic reactions to be beneficial for all 30 carbon conditions, ...". The authors have not clearly specified how this prediction was made. Is it predicted by EN and MLP or it's inferred by the SHAP interpretation? The authors need to provide more details, with Figure 2C which seems to suggest that this is an overall consensus number.

Reviewer #2:

Summary

This article used two machine learning methods, EN and MLP, to learn from gene-deletion data and predict reactions that are beneficial or detrimental to cell growth. This article innovatively addressed the explainability of traditional MLP models by introducing SHAP value calculation. An independent test sets qualitatively verified the trained model correctness. By consulting KEGG, the authors fully characterized both sets of beneficial and detrimental reactions in terms of their functions in the central metabolism. The authors also confirmed the model accuracy with knockout experiments. In general, this is a solid study that presents rigorous modeling and carefully designed experiments.

Review comments

1. "Regarding the critical reactions reported in the training data [4, 23], the EN model showed an average accuracy of 74.9% per carbon source"

Can the author provide a definition of "critical reactions"?

2. "These values were approximately 10% higher than those obtained when using the training data [4, 23], indicating that the constructed models did not overfit the training dataset."

This conclusion may not be well-founded. In the model training code, there is cross validation, so overfitting is avoided during model training. In general, it is very rare that testing data has better prediction results than the training data. Here when comparing training and testing performance, it is important to keep in mind that the training and testing datasets were from two different distributions. They may have different inherent conditions, such as growth conditions and strain genetics. The authors should revise this statement.

3. "However, the majority of metabolic reactions in the biosynthetic pathways were predicted to be beneficial for all carbon sources."

Do the authors mean biomass synthesis pathways here? Can the authors be specific about what biosynthetic pathways they are referring to here?

4. "These analyses demonstrated that biosynthetic pathways tend to be beneficial regardless of the carbon source, whereas beneficiality of energy-generating degradative pathways can vary depending on the specific carbon source"

Does the explainability of the ML models offer some reasons for this? It is specific to carbon source. Are there any patterns related to carbon sources' properties, such as Gibbs free energy or entering pathway?

5. Section "Experimental validation of model predictions beyond experimental training data"

Although ML is good at generalizing patterns in the training data, it should be noted that extrapolation beyond the training data may have some complications, especially in the highly connected central metabolism. The seven genes *cysE*, *glnA*, *hisA*, *metB*, *proA*, *purL*, and *serB* are mostly amino acids synthesis reactions. Does the model offer some explanation to justify this extrapolation?

6. In discussion section, "This study leveraged the power of ML and DL to predict the impact of metabolic reactions bacterial growth."

DL should be a subset of ML.

7. "The majority of the reactions involved in catabolic pathways, especially central carbon metabolism, were predicted to be beneficial or detrimental to cell growth, depending on the carbon source utilized (Figures 3 and 4), indicating that the type of carbon source can have an impact on the benefits of these pathways."

Does this shift in beneficial/detrimental reactions related to carbon sources' chemical or physical properties?

8. The detrimental reactions in carbohydrate reactions are intriguing. Does the model offer some explanations?

9. After GEM simulation, will taking the absolute value of each flux value lose some information about the reversible reactions?

Reviewer #3:

Manuscript Reference: MSB-2023-11953

Title: Machine learning identifies key metabolic reactions in bacterial growth on different carbon sources
Authors: Hyunjae Woo, Youngshin Kim, Dohyeon Kim and Sung Ho Yoon

This paper is a challenge to understand the effects of 30 different sugar sources on cell growth and their mechanisms down to the individual enzyme steps. Machine learning, especially deep learning, has been considered a black-box problem because of the difficulty in interpreting the results obtained from the learning process. In contrast, machine learning with constraint-based metabolic modeling (CBM) has opened the way to genotype-phenotype identification.

The authors constructed a model integrating genome-wide gene deletion data from two types of machine learning and deep learning with metabolic flux profiles from the growth data of the comprehensive deletion strains on a minimal medium with 30 different carbon sources. They compared them to identify enzyme steps that act as beneficial or detrimental. The identification of the enzymes and genes responsible for influencing cell growth in an intracellular metabolic network is a very difficult problem due to the complexity of the network structure. Previous analyses have only identified the essential enzymes, but this analysis shows that it is possible to identify the steps that promote non-essential growth in addition to the essential enzymes.

The experimental data were modeled and analyzed using data from Brown, ED. group's comprehensive analysis of 30 different carbon sources.

The predictions obtained from these analyses were validated in biological experiments using the single deletion or double

deletion strains.

The content of this paper goes into the problems of machine learning and develops a method to identify the causative gene (enzyme) by a method that enables interpretation of the results obtained from the machine learning and deep learning methods and experimentally confirmed the results, which will benefit readers of this journal. It is worth to be published.

I need, however, to request the authors a satisfactory explanation of the following points.

Major points:

- 1) An important aspect of the present analysis is that it allows us to approach the reasons why the results of previous machine learning analyses have been difficult to interpret. While it is possible to identify the genes responsible for the effects of individual carbon sources on growth, it is also possible to quantitatively seek the identification of the responsible genes, probably because the data from the comprehensive deletion strains have been modeled. I would like to request a brief explanation of the principle of the method so that readers in the biological field can easily understand the point.
- 2) Overall, the figure legends are insufficient, or the order of the figures needs to be considered. For example, the meaning of the coloring of each carbon source in Fig. 1C is explained in Fig. 2.
- 3) The results of the deletion of *tpiA* in Fig. 5B are noteworthy for the peculiar behavior of the case in which xylose was used as the carbon source. There is no mention of the possible cause of the difference between the predicted results from the model and the verification from biological experiments. This point needs to be explained in detail.
- 4) In Fig. 6A and D, the experimental validation of the deletion strains of *purN* and *purT* is shown. Why did the authors not perform the experimental verification of the *purN-purT* double deletion strain? If not, should they cite the reports of analysis of double deletion strains already available? (Nygaard and Smith, *J Bacteriol* 1993 Vol. 175, Pages 3591-7)

Minor points

- 1) Figure 4B is difficult to see. It may be easy to understand for researchers familiar with enzyme names, but it is difficult for those who are used to gene names to see. Please consider providing a separate figure with the gene names as supplementary information to make the figure simpler and easier to see.
- 2) Fig. 6A, B D, the flow from Alanin and acetate is depicted as a reference to the delay in growth with L-Alanin and why it cannot grow with acetate, but there is no mention of fatty acid. I assume that fatty acid also enters acetyl-CoA, but I have not been able to confirm if that is the only way. It is possible to infer that from the model, but I am unsure if it is the only one. Also, there is no explanation of the difference between the solid and dashed arrows in this figure.
- 3) Supplementary Table S4: In the column of Strain, JW ID is used to indicate the gene that is deleted, but it is considered appropriate to use ECK ID rather than JW.
- 4) Since there are many abbreviations, it would be easier to note them all together.

Point-by-point response to referees' comments

The authors deeply appreciate all the comments from the reviewers. We have carefully read through all of the reviews and have addressed all critiques.

< Reviewer 1 >

Hyunjae et al. propose machine learning models that integrate genome-wide gene-deletion data and simulated flux distributions to identify metabolic reactions beneficial or detrimental to *Escherichia coli* grown on 30 different carbon sources. The authors use FBA to generate fluxes. They interpret the models using SHAP and experimentally validate the metabolic reactions. Although the performance is reasonable even if fairly baseline machine learning models have been used, the way in which the features have been generated is not satisfactory, and this is likely to have strong effects on the accuracy that can be achieved by all models.

1) "Simulate oxaloacetate utilization, three metabolic reactions (BiGG ID: EX_oaa_e, OAA2_2pp, and OAAtex) were added to the model." Can the authors justify these manual curation steps, maybe adding key references showing that these three reactions are required for oxaloacetate?

Response: We thank for the reviewer's comment. For oxaloacetate uptake, we have modified the related sentence and added a reference as follows.

Modified sentence:

To allow the GEM to simulate oxaloacetate utilization, we added oxaloacetate (oaa) and three metabolic reactions (BiGG ID: EX_oaa_e, OAA2_2pp, and OAAtex) contained in *E. coli* K-12 DH10B GEM (iECDH10B_1368) (Monk *et al*, 2013) to iML1515.

2) "The maximum uptake rates of the carbon source and oxygen were set at 10 and 20 mmol/gDCW/h, respectively" The authors have not clarified how and why they constrain the carbon and oxygen uptake rates to 10 and 20 mmol/gDW/h. Are these values from the experimental data or from a previous study? Furthermore, the models have not been constrained using relevant media composition or transcriptomics data. Constraining the model is likely to help generate much more accurate features.

Response: We set the maximum uptake rates of the carbon source and oxygen as 10 and 20 mmol/gDW/h, respectively, because these values have been frequently used in FBA simulations for different carbon source utilization, especially in the studies of iML1515 reconstruction (Monk *et al*, 2017) and training dataset (Tong *et al*, 2020). In "MATERIALS AND METHODS" section of the submission, we had denoted that FBA simulations were performed using *in silico* MOPS minimal medium along with the constraints. We agree that integrating transcriptome data into metabolic simulation can improve the accuracy of the flux predictions. However, the availability of transcriptome data for specific carbon conditions is limited, especially when it comes to gene deletion mutants. We have revised the manuscript as follows.

Modified sentences:

The maximum uptake rates of the carbon source and oxygen were set at 10 and 20 mmol/gDCW/h, respectively (Monk *et al.*, 2017; Tong *et al.*, 2020).

To generate the input data, we performed gene deletion simulations using the genome-scale metabolic model (GEM) of *E. coli* K-12 MG1655 (iML1515) (Monk *et al.*, 2017) and *in silico* MOPS minimal media supplemented with each of the 30 carbon sources (Tong *et al.*, 2020).

Added Supplementary Table:

Appendix Table S4. Formulation of *in silico* MOPS minimal media. The maximum uptake rates of each carbon source and oxygen are set at 10 and 20 mmol/gDCW/h, respectively (Monk *et al.*, 2017; Tong *et al.*, 2020). Uptake rates of other components are unbounded.

Metabolite	Reaction ID	Reaction	Lower Bound	Upper Bound
Ammonia	EX_nh4_e	nh4[e] <=>	-1000	1000
Calcium	EX_ca2_e	ca2[e] <=>	-1000	1000
Chloride	EX_cl_e	cl[e] <=>	-1000	1000
Cobalt	EX_cobalt2_e	cobalt2[e] <=>	-1000	1000
Copper	EX_cu2_e	cu2[e] <=>	-1000	1000
ferrous ion	EX_fe2_e	fe2[e] <=>	-1000	1000
Magnesium	EX_mg2_e	mg2[e] <=>	-1000	1000
Manganese	EX_mn2_e	mn2[e] <=>	-1000	1000
Oxygen	EX_o2_e	o2[e] <=>	-20	1000
Phosphate	EX_pi_e	pi[e] <=>	-1000	1000
Potassium	EX_k_e	k[e] <=>	-1000	1000
Sodium	EX_na1_e	na1[e] <=>	-1000	1000
Sulfate	EX_so4_e	so4[e] <=>	-1000	1000
Zinc	EX_zn2_e	zn2[e] <=>	-1000	1000
Carbon source*	Ex_carbon_e	carbon[e] <=>	-10	1000

*Carbon source (exchange reaction): acetate (EX_ac_e), adenosine (EX_adn_e), D-alanine (EX_ala_D_e), fructose (EX_fru_e), fucose (EX_fuc_L_e), fumarate (EX_fum_e), galactose (EX_gal_e), galacturonate (EX_galur_e), gluconate (EX_glc_n_e), glucose (EX_glc_D_e), glucosamine (EX_gam_e), glucuronate (EX_glc_r_e), glycerol (EX_glyc_e), lactate (EX_lac_D_e), L-alanine (EX_ala_L_e), malate (EX_mal_L_e), maltose (EX_malt_e), mannitol (EX_mnl_e), mannose (EX_man_e), N-acetyl glucosamine (EX_acgam_e), oxaloacetate (EX_oaa_e), pyruvate (EX_pyr_e), ribose (EX_rib_D_e), saccharate (EX_glcr_e), sorbitol (EX_sbt_D_e), succinate (EX_succ_e), trehalose (EX_tre_e), thymidine (EX_thymd_e), xylose (EX_xyl_D_e), α -ketoglutarate (EX_akg_e).

3) The authors simulated the metabolic fluxes using FBA and MOMA, which were used as input features for machine learning and deep learning models. However, the flux rates generated by FBA for most of the reactions are zeros, which are not considered good features for machine learning models. Therefore, it is recommended to use FVA or flux sampling techniques (<https://www.ncbi.nlm.nih.gov/pmc/articles/PMC6718391/>) to generate more accurate features for machine learning models.

Response: In this study, MOMA was used to simulate the metabolic flux distribution of the mutant strain. MOMA is an algorithm specifically designed to predict the behavior of a mutant strain. FVA and flux sampling aim to explore the potential range of flux values for every metabolic reaction in a

metabolic network, instead of calculating a single optimal solution. FVA is used to determine the minimum and maximum allowable flux values for each reaction in a metabolic network. Flux sampling is a stochastic approach used to explore the space of possible flux distributions in a metabolic network. In general, they, especially flux sampling, generate non-zero flux values more than MOMA. However, in machine learning, values of each feature in each input data need to be specified to prepare and structure the data for training and prediction. In other words, each data point needs to have a well-defined set of feature values. In this sense, we think that FVA and flux sampling are not appropriate for preparing the input data, because they gave multiple flux values for each metabolic reaction (feature) in each input data. We have revised the manuscript as follows.

Modified sentence:

To identify the metabolic responses affecting cell growth, predictive models were constructed using supervised learning algorithms that mapped metabolic fluxes derived from **minimization of metabolic adjustment (MOMA) simulations** as input data to experimental growth data as output data (Fig 1A).

Sentence added to the “Preparation of training dataset for supervised learning” section:

We also evaluated a different type of flux simulation methods such as flux variability analysis (Gudmundsson & Thiele, 2010) and flux sampling (Herrmann *et al*, 2019), which can generate non-zero flux values for a wide range of reactions in Appendix Text 1 and Appendix Fig S1.

Added Appendix Text:

Appendix Text 1: Evaluation of flux variability analysis and flux sampling as a flux simulation method

In machine learning, values of each feature in each input data need to be specified to prepare and structure the data for training and prediction. In this study, MOMA, an algorithm specifically designed to predict the behavior of a mutant strain (Segrè *et al*, 2002), was used to simulate the metabolic flux distribution of the mutant strain under different carbon conditions. Flux variability analysis (FVA) (Mahadevan & Schilling, 2003) and flux sampling (Herrmann *et al.*, 2019) can be alternative methods for flux simulation, offering the advantage of generating non-zero flux values for a larger number of reactions when compared to MOMA. To address this, we applied FVA and flux sampling to generate flux values for glucose condition, by setting the maximum uptake rates of the glucose and oxygen at 10 and 20 mmol/gDCW/h, respectively. Among 2,715 reactions, MOMA generated non-zero flux values for every 435 reactions. FVA yielded non-zero flux values for 565 reactions with at least one non-zero value at the minimum and maximum flux values (Appendix Fig S1A). Within the FVA results, 61 reactions had extremely low minimum flux values (28 ea) (< -990 mmol/gDCW/h), and/or extremely high maximum flux values (44 ea) (> 995 mmol/gDCW/h). Except for these reactions with unrealistic flux values, flux values from MOMA were in good agreement with the minimum flux values from FVA for most reactions (97.8%, 2654 ea) ($r = 0.96$). Flux sampling generated a nonzero flux distribution for every 1,786 reactions without producing any negative flux values (Appendix Fig S1B). Within the results from flux sampling, 61 reactions had extremely high minimum flux values (26 ea) (> 136 mmol/gDCW/h), and/or extremely high maximum flux values (61 ea) (> 104 mmol/gDCW/h). Even when excluding these reactions with unrealistic flux values, most flux values from MOMA (97.8%, 2654

reactions) showed poor agreement with the minimum ($r = 0.22$) or average ($r = 0.28$) flux values obtained from flux sampling.

FVA and flux sampling are designed to capture the potential range of flux values for every metabolic reaction in a metabolic network, rather than finding a single optimal solution. Additionally, as illustrated in Appendix Fig S1, they generated unrealistic flux values for many reactions which could introduce noise or distortions when using this data for machine learning applications. Considering the importance of well-defined and reliable feature values for each data point in machine learning, there's a need to develop new algorithms that enable FVA or flux sampling to generate flux data suitable for machine learning applications.

Added Appendix Figure:

Appendix Figure S1: Comparison of flux distributions obtained from the flux simulations using MOMA (metabolic simulation of minimization of metabolic adjustment), FVA (flux variance analysis), and flux sampling. To simulate growth using *in silico* MOPS minimal medium supplemented glucose as the sole carbon source, the maximum uptake rates of the carbon source and oxygen were set at 10 and 20 mmol/gDCW/h, respectively. FVA and flux sampling were performed using the "flux_variability_analysis" function (with default setting) and "OptGPSampler" function (with default setting of 100 iterations), of the COBRApy software (Ebrahim *et al*, 2013), respectively.

- A. Comparison of flux values from FVA and MOMA simulations. For each of the 565 metabolic reactions with at least one non-zero value at the minimum and maximum flux values in the FVA simulation, the flux range (from minimum to maximum fluxes) in the FVA simulation was compared with the flux value in the MOMA simulation. The y-axis denotes the flux values.
- B. Comparison of flux values from flux sampling and MOMA simulations. For each of the 1,786 metabolic reactions with as a nonzero flux distribution, the flux distribution in the simulation of flux sampling was compared with the flux value in the MOMA simulation. The left y-axis denotes the flux distributions from the flux sampling simulation as mean \pm standard deviation on a log10 scale. The right y-axis indicates flux values from the MOMA simulation.

4) "Thus, we assessed model accuracy by comparing the reactions (features) selected by the models with the output data of the training dataset [4, 23] ...". The authors have designed regression models however, they have not clearly mentioned how the accuracy of the model was evaluated. The authors need to provide more details (mathematically would be best) on how exactly the models'

accuracy was evaluated.

Response: We measured accuracy of each model by counting how many experimentally determined critical reactions were contained in the list of positive reactions predicted by the model. To clarify how the model accuracy was calculated, we have revised the manuscript as follows.

Modified paragraph:

It can be speculated that the removal of beneficial metabolic reactions reduces the final biomass or even prevents cell growth. Regarding two datasets for phenotypic screening of single-gene deletions: one being the training dataset (Baba *et al.*, 2006; Tong *et al.*, 2020) and the other an independent dataset (Monk *et al.*, 2017), critical reactions within these datasets were obtained from the respective papers that reported these datasets. The accuracy of each model was evaluated by counting the number of critical reactions found among the positive reactions predicted by the model (Fig 2B and Table EV2; Materials and Methods).

Paragraph added to the MATERIALS AND METHODS section:

Measurement of model accuracy

Accuracy of EN and MLP models were separately measured against two datasets of the training dataset (Baba *et al.*, 2006; Tong *et al.*, 2020) and an independent dataset (Monk *et al.*, 2017). In the studies reporting the datasets (Monk *et al.*, 2017; Tong *et al.*, 2020), the authors defined critical genes by their own criteria. To retrieve critical reactions in each dataset, those reported critical genes (Table EV2) were mapped to the corresponding metabolic reactions using the gene- protein-reaction (GPR) rules included in the iML1515 model (Monk *et al.*, 2017). As the critical reactions were determined based on the phenotypic screen of the single-gene deletions, each reaction is highly likely to be directly influenced by the absence of a specific gene. Thus, for a fair comparison, among the beneficial reactions predicted by each model, we only considered metabolic reactions that are mediated by a single protein or protein complex and are not otherwise activated by other proteins. The prediction accuracy of each model against a specific dataset was calculated as follows:

$$\text{Prediction accuracy} = P(B/A) = \frac{\text{number of } (A \cap B)}{\text{number of } A},$$

where A is the beneficial reactions predicted by a model, and B is the critical reactions reported in an experimental dataset.

5) There seems to be an extremely high discordance between the EN and the MLP approach when predicting the detrimental steps. Given that 7 reactions are predicted under all carbon sources, is it because the EN method is too permissive when selecting key features, or is the MLP too strict?

Response: The discrepancy in detrimental reactions predicted by EN and MLP models can be attributed to the inherent characteristics of the output data and the feature selection methods. To address the comment, we have modified the manuscript as follows.

Paragraph added to the DISCUSSION section:

The high discordance between the EN and MLP models was observed for predicting detrimental reactions, especially when seven reactions were predicted only by EN model to be detrimental under all carbon sources. This can be attributed to the inherent characteristics of the output data and the feature selection methods. The training output dataset examining cell growth of gene deletion mutants (Baba *et al.*, 2006; Tong *et al.*, 2020), displayed much more occurrences of reduced biomasses than increased ones. Moreover, the extent of biomass reduction, sometimes even resulting in no growth, was not only more pronounced but also displayed a wider variability than the cases of growth enhancement. This data bias towards reduced cell growth can lead the MLP model to assign high positive SHAP values to more reactions. As SHAP values are calculated within the context of feature interactions and dependencies (Lundberg & Lee, 2017), features with such highly positive SHAP values might dominate, potentially overshadowing or constraining the influence of features with negative SHAP values. On the other hand, the regression coefficients in the EN model aren't heavily influenced by feature interactions, which might make it better suited to predict detrimental reactions compared to the MLP model.

6) "Regarding the critical reactions reported in the training data [4, 23], the EN model showed an average accuracy of 74.9% per carbon source,...". A machine learning model with an average accuracy of 74.9% is not sufficient to convince that the proposed framework (CBM with machine learning) is the best to identify the metabolic reactions that significantly influence bacterial cell growth.

Response: The seemingly insufficient accuracy was caused by the presence of the predicted beneficial reactions that did not agree with the experimental dataset. For instance, while both EN and MLP models predicted 22 metabolic reactions as beneficial under glucose conditions, yet the experimental training data showed no reduced cell growth upon their removal. However, additional growth experiments validated defected or reduced growth in 19 of the reaction deletions. Considering these validated predictions, the accuracy of the EN and MLP models for the glucose condition, increased from initial estimates of 80.7% and 78.7% in model accuracy to 96.6% and 93.7%, respectively. To clarify this, we have modified the "Experimental validation of model predictions beyond experimental training data" section and moved it directly below the "Model accuracy" section. We also repositioned the previous Figure 5 to be Figure 2C (Please see our response to Reviewer 3's #2).

Modified paragraph:

Notably, the model predictions included certain cases that **were inconsistent with** the experimental training data. For example, both EN and MLP models predicted 22 metabolic reactions **as** beneficial under glucose conditions; however, **the** experimental training data showed that **removal of each of them** did not reduce cell growth on glucose (Tong *et al.*, 2020). The models correctly predicted the essentiality of three genes (*coaA*, *coaE*, and *hemE*) that were initially thought to be non-essential (Baba *et al.*, 2006), but later found to be essential (Yamamoto *et al.*, 2009). In additional growth experiments, 16 of 19 gene deletions led to growth defects or reduced growth (Figure 2C). **Considering these validated predictions, the accuracy of the EN and MLP models for the glucose condition, increased from initial estimates of 80.7% and 78.7% in model accuracy to 96.6% and 93.7%, respectively. This demonstrates that our models are robust to experimental uncertainty in the training dataset.**

7) Given the final results on the accuracy, a thorough examination of the main contribution presented by the paper is missing. For instance, it has not been investigated whether features could

have been generated in a different way. The authors could have tried generating more accurate features (for instance, via FVA or flux sampling techniques), then applying feature selection techniques (e.g., variance threshold, tree-based feature selection) to reduce high dimensional data and remove noise (irrelevant features). Most importantly, the authors need to clarify how the accuracy of the regression models was evaluated, as this would elucidate likely steps that could improve it.

Response: In our responses to Reviewer 1's comments #3, 4 and 6 regarding model accuracy and feature generation, we believe that this study represents rigorous modeling that contributes to understanding the effects of different carbon sources on cell growth and their mechanisms down to the individual metabolic steps. When preparing the input dataset for supervised learning for each carbon condition, we removed features that had zero values in all mutation simulations to reduce the high dimensionality of the data. This can be seen as a form of variance thresholding where the threshold is set to zero, since all fluxes with constant values had a value of zero, with only one exception of the flux for the ATP maintenance requirement. To address the comment, we modified the manuscript as follows.

Modified sentence:

Then, we removed the features that had zero values across all 1,422 mutation simulations to reduce the high dimensionality of the data. This can be seen as a form of variance thresholding (Fida *et al.*, 2021) where the threshold is set to zero, since all fluxes with constant values had a value of zero, except for only one reaction of the ATP maintenance requirement. We also evaluated two widely used techniques for feature selection, the variance thresholding and random forest (Fida *et al.*, 2021), in Appendix Text S2 and Appendix Table S1.

Added Appendix Text:

Appendix Text S2: Evaluation of the variance thresholding and random forest as a feature selection method

To increase the performance of machine learning algorithms, feature selection is an important data preprocessing step to reduce the dataset size by eliminating redundant and irrelevant features. The primary goal of feature selection is to identify the optimal subset of features without losing the key features of the data. However, this is an NP-hard (non-deterministic polynomial-time hard) problem (Kohavi & John, 1997), which led to the development of numerous techniques to tackle this issue (Venkatesh & Anuradha, 2019). We tested two commonly used methods of feature selection (Fida *et al.*, 2021): the variance thresholding and tree-based random forest on the glucose training dataset.

In our construction of ML models, we removed any features that consistently showed zero values across all mutation simulations. This can be seen as a form of variance thresholding (Fida *et al.*, 2021) with threshold set to zero, since all fluxes with constant values had a value of zero, except for only one reaction of the ATP maintenance requirement. For the glucose dataset, the number of selected features decreased from 2715 to 1914. When we implemented variance thresholding with a set threshold of zero, the number of selected features decreased from 2715 to 1913.

Compared to our ML models, variance thresholding with threshold set to one significantly reduced the number of features (from 2715 to 681), and also decreased the accuracy of the MLP model. Applying the random forest algorithm using `RandomForestRegressor` function with default settings in

the sklearn Python package led to only 123 selected features, and also decreased the accuracy of the EN model. Thus, features selected in this study can be considered a subset of features representing salient characteristics of the data, although they might not be the most optimal selection.

Appendix Table S1 Comparison of prediction accuracy of the ML models for the glucose condition using different feature selection methods as a preprocessing step.

	Variance thresholding				Random forest ^b	
	Threshold of 0 (This study)		Threshold of 1 ^b			
Feature reduction (ea, before -> after)	2715 -> 1914		2715 -> 681		2715 -> 123	
Model	EN	MLP	EN	MLP	EN	MLP
Beneficial reactions (ea)	291	351	81	188	36	56
Detrimental reactions (ea)	41	24	29	14	11	7
Model accuracy ^a	80.7%	78.7%	80.0%	50.0%	57.1%	77.8%

^aThe accuracy of each model was evaluated by counting the number of critical reactions found among the positive reactions predicted by the model. The critical reactions were obtained from the paper reporting the training dataset for the glucose condition (Tong *et al.*, 2020).

^bJupyter notebooks for these tasks are available on https://github.com/sybirg/xai_growth/tree/main/Supplementary.

8) "Supplementary Table S3" is missing.

Response: In the submission, Table S3 was provided as a Data Set in Excel format. We are sorry for the confusion. We have now uploaded Table S3 as Table EV2.

9) In the deep learning pipeline, the authors tuned the hyperparameters first and used the same hyperparameters while training the model for ten seeds (iterations). However, in this case, this is not the best way to tune the hyperparameters. It is recommended to tune the hyperparameters for each iteration. While training the model with a smaller sample size, it is best recommended to train the models using nested k-fold validation where the inner fold is used to tune the hyperparameter and train the model with the best hyperparameter and the outer fold is used to evaluate the model (<https://www.mdpi.com/2076-3417/12/13/6681>).

Response: The optimal strategy for hyperparameter tuning in a setting with multiple seeds (iterations) is often debated in machine learning applications. The decision to tune hyperparameters once versus in each iteration depends on the particular objectives of the study, the available computational resources, and the desired balance between performance optimization and model generalization. In this study, when training the MLP model, using a fixed set of hyperparameters across 10 different seeds led to equivalently favorable results, demonstrated by both the training loss and validation loss converging to zero. This indicates a stable model and the chosen hyperparameters tailored for this specific problem and dataset. To reflect this, we have modified the manuscript as follows.

Sentence added to the RESULT section:

The MLP model for each carbon condition was trained using a predetermined set of hyperparameters, and interpreted with SHAP across ten iterations. When training the MLP model, both the training loss

and validation loss converged to zero (https://github.com/sybirg/xai_growth/tree/main/Supplementary), indicating that the model is stable and the chosen hyperparameters were well-suited for this specific problem and dataset.

Plots added to https://github.com/sybirg/xai_growth/tree/main/Supplementary (Two of 30 subfigures are shown here):

Training and validation loss of MLP models. The total training and validation loss over epochs obtained by training MLP models across 30 different carbon dataset each with 10 different random seeds are shown.

10) The code could not be checked in full. The "deep_learning.ipynb" script throws an exception at line:

`X_train_scaled, y_train = sklearn.utils.shuffle(X_train_scaled, y_train, random_state=random_seed).`
It seems that the author missed defining the X_train_scaled and y_train variables.

Response: We appreciate the reviewer’s effort in reviewing the code. We have thoroughly revised the code on GitHub. Regarding the comment, we have added the following code.

Code added to the file “deep_learning.ipynb” on GitHub https://github.com/sybirg/xai_growth:

```
X_train_scaled = sklearn.preprocessing.StandardScaler().fit_transform(X_data)
y_train = y_data
```

Update of codes on Zenodo:

We have released an updated version (v2) of our Python codes on Zenodo, with a DOI of <https://zenodo.org/records/10164986>. This update includes details necessary for reproducing the results presented.

11) "The MLP model predicted 301 metabolic reactions to be beneficial for all 30 carbon conditions, ...". The authors have not clearly specified how this prediction was made. Is it predicted by EN and MLP or it's inferred by the SHAP interpretation? The authors need to provide more details, with Figure 2C which seems to suggest that this is an overall consensus number.

Response: For clarity, we have modified the related sentence as follows.

Modified sentence:

Based on SHAP interpretation, the MLP model predicted 301 metabolic reactions to be beneficial for all 30 carbon conditions. Remarkably, these MLP predictions included all of the 221 reactions predicted by the EN model as well as all of the 214 essential reactions predicted by the single-reaction deletion simulation.

< Reviewer 2 >

This article used two machine learning methods, EN and MLP, to learn from gene-deletion data and predict reactions that are beneficial or detrimental to cell growth. This article innovatively addressed the explainability of traditional MLP models by introducing SHAP value calculation. An independent test sets qualitatively verified the trained model correctness. By consulting KEGG, the authors fully characterized both sets of beneficial and detrimental reactions in terms of their functions in the central metabolism. The authors also confirmed the model accuracy with knockout experiments. In general, this is a solid study that presents rigorous modeling and carefully designed experiments.

Review comments

1) "Regarding the critical reactions reported in the training data [4, 23], the EN model showed an average accuracy of 74.9% per carbon source" Can the author provide a definition of "critical reactions"?

Response: We thank for the reviewer's comment. The critical genes in the datasets were obtained from the papers reporting the datasets. Regarding the critical genes and model accuracy, please see our response to Reviewer 1's comment #4.

2) "These values were approximately 10% higher than those obtained when using the training data [4, 23], indicating that the constructed models did not overfit the training dataset." This conclusion may not be well-founded. In the model training code, there is cross validation, so overfitting is avoided during model training. In general, it is very rare that testing data has better prediction results than the training data. Here when comparing training and testing performance, it is important to keep in mind that the training and testing datasets were from two different

distributions. They may have different inherent conditions, such as growth conditions and strain genetics. The authors should revise this statement.

Response: Thank you for your insightful observation. We agree that machine learning models are expected to perform well on the data they were trained on, compared to the unseen independent data. As mentioned earlier, we measured accuracy of each model by counting how many experimentally determined critical reactions were contained in the list of positive reactions predicted by the model. The critical gene lists were obtained from their respective studies, leading to distinct sets of critical genes in the training data and the independent additional data. This discrepancy could arise from various factors including differences in decision criteria, culture media, and cultivation time. Consequently, the level of experimental noise may vary across the datasets. Although this aspect requires experimental validation, we believe that higher concordance between model predictions and the additional dataset demonstrates a favorable feature regarding the model's generalization capability. To reflect this, we have modified the sentence as follows.

Modified sentence:

These values were approximately 10% higher than those obtained when using the training data (Baba *et al.*, 2006; Tong *et al.*, 2020), suggesting that the constructed models have generalized well beyond the training data.

3) "However, the majority of metabolic reactions in the biosynthetic pathways were predicted to be beneficial for all carbon sources." Do the authors mean biomass synthesis pathways here? Can the authors be specific about what biosynthetic pathways they are referring to here?

Response: The biosynthetic pathways here mean those for amino acids and purine, and other metabolites as shown in Figure 4A (red arrows). For clarity, we have modified the related sentence as follows.

Modified sentence:

However, the majority of metabolic reactions involved in the biosynthetic pathways were predicted to be beneficial for all carbon sources. These include pathways for synthesizing amino acids, purines, pyrimidines, and lipopolysaccharides.

4) "These analyses demonstrated that biosynthetic pathways tend to be beneficial regardless of the carbon source, whereas beneficiality of energy-generating degradative pathways can vary depending on the specific carbon source" Does the explainability of the ML models offer some reasons for this? It is specific to carbon source. Are there any patterns related to carbon sources' properties, such as Gibbs free energy or entering pathway?

Response: Such discrepancy in model predictions can be attributed to the way these pathways are structured and positioned within the metabolic network. We have modified the related discussion as follows.

Modified paragraph in DISCUSSION section:

The ML models predicted that the majority of reactions involved in biosynthetic pathways, especially in amino acid synthetic pathways, are beneficial to cell growth regardless of the carbon source, while involved in catabolic pathways, especially in central carbon metabolism, became beneficial or detrimental, depending on the carbon source utilized (Figures 3 and 4). This discrepancy can be attributed to how these pathways are structured and how the carbon source is mainly catabolized through the metabolic network. Catabolic pathways typically are highly interconnected, offering multiple alternative routes for carbon source utilization (Kim & Copley, 2007). For example, the upper EMP pathway was predicted to be beneficial for carbon sources such as glucose, *N*-acetyl glucosamine, fructose that enter through that pathway. However, for carbon sources like acetate, α -ketoglutarate, which follow different catabolic routes, EMP pathway was not predicted to be beneficial. In contrast, biosynthetic pathways tend to be linear and require activation in the absence of externally supplied amino acids or other essential building blocks. This interpretation of the model predictions underlines the importance of the structural characteristics of metabolic pathways in understanding how microorganisms respond to nutritional changes, requiring further in-depth investigation to unravel the exact underlying mechanisms.

5) Section "Experimental validation of model predictions beyond experimental training data" Although ML is good at generalizing patterns in the training data, it should be noted that extrapolation beyond the training data may have some complications, especially in the highly connected central metabolism. The seven genes *cysE*, *glnA*, *hisA*, *metB*, *proA*, *purL*, and *serB* are mostly amino acids synthesis reactions. Does the model offer some explanation to justify this extrapolation?

Response: In the submission, we meant "model predictions beyond experimental training data" by predicting essentiality of genes whose deletions were not tested in the output data of the training dataset. Despite being beyond the scope of the experimental output data, the feature fluxes associated with these genes are still included in the simulated input data, providing necessary information for the models to make predictions about those genes' essentiality. This scenario should fit the concept of interpolation rather than extrapolation. To avoid the possible misinterpretation, we have modified the manuscript. Those seven genes predicted to be essential under all carbon conditions are mostly involved in amino acids biosynthesis, probably because biosynthetic pathways tend to be linear and require activation in the absence of externally supplied amino acids or other essential building blocks. (Please see our response to Reviewer 2's comment #4.)

Modified sentence:

Our models predicted the essentiality of genes whose deletions were not tested in the output data of the training dataset (Fig 2D). Despite being beyond the scope of the experimental output data, the feature fluxes associated with these genes are still included in the simulated input data, providing necessary information for the models to make predictions about those genes' essentiality. Thus, these cases should fit the concept of interpolation rather than extrapolation.

6) In discussion section, "This study leveraged the power of ML and DL to predict the impact of metabolic reactions bacterial growth." DL should be a subset of ML.

Response: Thank you for the correction. We have modified the related sentences as follows.

Modified sentences:

This study leveraged the power of **two ML methods, EN and MLP**, to predict the impact of metabolic reactions bacterial growth.

Here, **the elastic net model and multilayer perception model** that integrated genome-wide gene-deletion data and simulated flux distributions were constructed to identify metabolic reactions beneficial or detrimental to *Escherichia coli* grown on 30 different carbon sources.

Experimental growths of single-gene deletions and metabolic simulations were combined to train **two explainable ML models**.

7) "The majority of the reactions involved in catabolic pathways, especially central carbon metabolism, were predicted to be beneficial or detrimental to cell growth, depending on the carbon source utilized (Figures 3 and 4), indicating that the type of carbon source can have an impact on the benefits of these pathways." Does this shift in beneficial/detrimental reactions related to carbon sources' chemical or physical properties?

Response: We are not sure that such shift is related to carbon sources' chemical or physical properties, as metabolic modelling only considers reaction stoichiometry of a metabolic network. Rather, we think that the varied beneficiality of reactions according to the carbon source can be attributed to the how these pathways are structured and how the carbon source is mainly catabolized through the metabolic network. Regarding this, please see our response to Reviewer 2's comment #4. In addition to this, we have provided new Supplementary Figures showing the mapping of the model predictions onto metabolic pathways according to each of 30 carbon conditions.

Added Supplementary Figure (Two of 30 subfigures are shown here):

Appendix Figure S2. Mapping of the model prediction onto metabolic pathways for each of 30 carbon source conditions.

The solid arrows indicate single metabolic reactions, and dashed arrows indicate multiple sequential reaction steps. The color scheme of each arrow indicates the impact of each metabolic reaction on cell growth: red, beneficial; blue, detrimental; black, not significant.

Glucose

Acetate

8) The detrimental reactions in carbohydrate reactions are intriguing. Does the model offer some explanations?

Response: In the section “Experimental validation of metabolic reactions predicted to be detrimental to cell growth” in the submission, we had provided some extent of explanation for the validated detrimental reactions. However, model’s ability to explain precise cellular and molecular mechanisms behind model predictions is limited, which we feel beyond this study. To reflect this, we have modified the related sentences as follows.

Modified sentence:

The predictive models developed in this study provide insights into **which** metabolic reactions influence cell growth in the presence of different carbon sources. **Although their ability to explain the underlying metabolic and regulatory mechanisms is limited, the intriguing predictions, including the detrimental reactions in central catabolic pathways, serve as hypotheses that guide further experimental investigations,** enhancing our knowledge of cellular metabolism and its role in cell growth.

9) After GEM simulation, will taking the absolute value of each flux value lose some information about the reversible reactions?

Response: Taking the absolute value of each flux value lose information on directionality of reversible reactions. Nonetheless, we did this because ML models are intrinsically unable to understand the biochemical meaning behind negative flux values representing reverse reactions. When ML models are trained, negative flux values are misinterpreted as representing low activity of reactions rather than reverse reactions, potentially leading to incorrect predictions, such as identifying these reactions as detrimental to cell growth. To avoid this, splitting a reversible reaction into separate forward and reverse reactions can be considered. However, this approach will result in a high dimensional data, which introduce artifacts or hinder model training due to the curse of dimensionality and increased computational demands. To address this, we have modified the related sentence as follows.

Modified sentence:

We took the absolute value of each flux **to prevent the misinterpretation of negative flux values as representing low reaction activity rather than reverse reactions, during the model training.**

< Reviewer 3 >

This paper is a challenge to understand the effects of 30 different sugar sources on cell growth and their mechanisms down to the individual enzyme steps. Machine learning, especially deep learning, has been considered a black-box problem because of the difficulty in interpreting the results obtained from the learning process. In contrast, machine learning with constraint-based metabolic modeling (CBM) has opened the way to genotype-phenotype identification.

The authors constructed a model integrating genome-wide gene deletion data from two types of machine learning and deep learning with metabolic flux profiles from the growth data of the comprehensive deletion strains on a minimal medium with 30 different carbon sources. They compared them to identify enzyme steps that act as beneficial or detrimental. The identification of the enzymes and genes responsible for influencing cell growth in an intracellular metabolic network

is a very difficult problem due to the complexity of the network structure. Previous analyses have only identified the essential enzymes, but this analysis shows that it is possible to identify the steps that promote non-essential growth in addition to the essential enzymes.

The experimental data were modeled and analyzed using data from Brown, ED. group's comprehensive analysis of 30 different carbon sources. The predictions obtained from these analyses were validated in biological experiments using the single deletion or double deletion strains. The content of this paper goes into the problems of machine learning and develops a method to identify the causative gene (enzyme) by a method that enables interpretation of the results obtained from the machine learning and deep learning methods and experimentally confirmed the results, which will benefit readers of this journal. It is worth to be published. I need, however, to request the authors a satisfactory explanation of the following points.

[Major points]

1) An important aspect of the present analysis is that it allows us to approach the reasons why the results of previous machine learning analyses have been difficult to interpret. While it is possible to identify the genes responsible for the effects of individual carbon sources on growth, it is also possible to quantitatively seek the identification of the responsible genes, probably because the data from the comprehensive deletion strains have been modeled. I would like to request a brief explanation of the principle of the method so that readers in the biological field can easily understand the point.

Response: We thank for the reviewer's suggestion. To explain the biological rationale behind the method, we have added a paragraph as follows.

Paragraph added to the DISCUSSION section:

In this study, ML models were developed to pinpoint metabolic reactions that markedly influence bacterial growth under different carbon source conditions. The basic rationale for the causal predictions is that the GEM underpinning flux simulations is mechanistically constructed; thus, the features constituting the trained ML models become mechanistically causal (Sahu *et al*, 2021; Yang *et al*, 2019). ML models learn patterns and relationships between input data and output, and whether the predictions are causal or correlative depends on the nature of the input data and the underlying relationships. If the input features are primarily observational (such as omics data), ML models typically identify correlations or associations between input features and the output variable. To make causal predictions (meaning that changing input features will directly cause changes in the output), input features need to represent causative factors for the output variable. A GEM is a mathematical representation of interconnected metabolic reactions and compounds, facilitating metabolic simulations of how an organism transforms nutrients into energy and other necessary molecules. Because the GEM-based fluxes as the input features are derived from the mechanistically constructed GEM, the ML models are expected to make causal predictions that reflect the direct influence of metabolic reactions on growth outcomes in different carbon conditions. The qualitative input-output relationships were learned and quantified by assigning specific numerical values, and we determined which input feature reactions had statistically significant relationships with the output data.

2) Overall, the figure legends are insufficient, or the order of the figures needs to be considered. For example, the meaning of the coloring of each carbon source in Fig. 1C is explained in Fig. 2.

Response: Thank you for the kind suggestion. To address this, we have revised the figures and figure legends as following. We have also changed the order of some figures.

Modified Figures:

Figure 1C: The coloring of each carbon source in the outer circle was removed, as it is unnecessary for explaining the overall scheme of this study.

Figure 2: The previous Figure 5 has been repositioned to be Figure 2C, as we have modified the “Experimental validation of model predictions beyond experimental training data” section and moved it directly below the “Model accuracy” section (Please see our response to Reviewer 1’s #6). The previous Figure 2C has now been presented as Fig EV2.

Figure EV2: The previous Figure 2C is now Figure EV2.

Figure 5 (previously Figure 6): Subpanel B and subpanel C have been swapped, and accordingly, the associated text paragraphs have been repositioned.

Modified Figure Legends: Figures 1, 2, 4, and 5

3) The results of the deletion of *tpiA* in Fig. 5B are noteworthy for the peculiar behavior of the case in which xylose was used as the carbon source. There is no mention of the possible cause of the difference between the predicted results from the model and the verification from biological experiments. This point needs to be explained in detail.

Response: Although $\Delta tpiA$ was not included in the training growth data, its inability to grow on solid agar media supplemented with xylose as the sole carbon source has been reported (Monk *et al.*, 2017). Moreover, considering the training data also used solid agar media, this prediction is hard to be considered a false positive. To reflect this, we have modified sentences and a Supplementary Figure showing bacterial growth on xylose as follows.

Modified sentence:

The deletion of *tpiA* encoding triosephosphate isomerase resulted in growth defects on the tested carbon sources, with xylose being the exception. Although $\Delta tpiA$ was not included in the training growth data, it was reported to be unable to grow on the agar plate containing minimal xylose medium (Monk *et al.*, 2017). Moreover, considering the training data also used solid agar media, this prediction is hard to be considered a false positive. The notable growth of $\Delta tpiA$ on xylose, despite an extended lag-time in liquid culture (Fig EV1), warrants further genetic and molecular investigations, as $\Delta tpiA$ has been reported to be incapable of growing on glucose and glycerol due to the accumulation of highly toxic methylglyoxal (Velur Selvamani *et al.*, 2014).

Added Supplementary Figure:

Figure EV1. Experimental validation of predicted beneficial reactions with missing deletions in the training data (Xylose condition).

The y-axis denotes relative biomasses of gene-deletion mutants grown in the MOPS minimal medium supplemented with xylose carbon for 48 h. Growth measurements are represented as the mean \pm SEM from three independent cultivations.

4) In Fig. 6A and D, the experimental validation of the deletion strains of *purN* and *purT* is shown. Why did the authors not perform the experimental verification of the *purN-purT* double deletion strain? If not, should they cite the reports of analysis of double deletion strains already available? (Nygaard and Smith, J Bacteriol 1993 Vol. 175, Pages 3591-7)

Response: To address the comment, we have added a sentence along with the corresponding reference as follows.

Added sentence:

We did not test the double deletion strain of *purN* and *purT* as it requires an exogenous purine source for growth (Nygaard & Smith, 1993).

[Minor points]

1) Figure 4B is difficult to see. It may be easy to understand for researchers familiar with enzyme names, but it is difficult for those who are used to gene names to see. Please consider providing a separate figure with the gene names as supplementary information to make the figure simpler and easier to see.

Response: To address the comment, we have added Figure EV3 displaying metabolic gene names.

2) Fig. 6A, B D, the flow from Alanin and acetate is depicted as a reference to the delay in growth with L-Alanin and why it cannot grow with acetate, but there is no mention of fatty acid. I assume that fatty acid also enters acetyl-CoA, but I have not been able to confirm if that is the only way. It is possible to infer that from the model, but I am unsure if it is the only one. Also, there is no explanation of the difference between the solid and dashed arrows in this figure.

Response : As the reviewer suggested, the lack of growth of the double-deletion mutant ($\Delta aceB\Delta glcB$) on fatty acids can be inferred from the model, considering that fatty acids are also degraded directly into acetyl-CoA via β -oxidation (Dolan & Welch, 2018). For example, oleic acid was reported to be degraded mostly by β -oxidation (90%), while approximately 10% of the oleic acid is only partially degraded to 3,5-*cis*-tetradecadienoyl-CoA (Nie *et al*, 2008). To test this prediction, we conducted a growth experiment with the $\Delta aceB\Delta glcB$, utilizing oleic acid as the sole carbon source, which led to no observable growth. To reflect these, we have modified the manuscript as follows.

Modified sentences:

Double-gene deletion resulted in no growth on acetate, and extended the lag time for D-alanine and L-alanine (Fig 5C). Considering that fatty acids are also degraded directly into acetyl-CoA, which then needs to be metabolized via the glyoxylate shunt (Dolan & Welch, 2018), growth defect of $\Delta aceB\Delta glcB$ on fatty acids can be inferred from the model. As expected, $\Delta aceB\Delta glcB$ did not grow in the minimal medium supplemented with oleic acid as the sole carbon source (Appendix Figure S3). The extended the lag time of $\Delta aceB\Delta glcB$ growing on D-alanine and L-alanine is intriguing because alanine is purely a glucogenic amino acid which are not degraded directly into acetyl-CoA.

Added Supplementary Figure:

Appendix Figure S3. Growth experiment of $\Delta aceB\Delta glcB$ using oleic acid as the sole carbon source. Using MOPS minimal media supplemented with 5mM oleic acid in the presence of 0.4% Triton X-100, $\Delta aceB\Delta glcB$ did not grow, in contrast to the cell growth of its parent wild-type *E. coli* K-12 BW25113. Error bars represent the standard error of the mean (SEM) from three independent cultivations.

Sentence added to the legend of Figure 5A (previously Figure 6A):

The solid arrows indicate single metabolic reactions, and dashed arrows represent multiple sequential reaction steps.

3) Supplementary Table S4: In the column of Strain, JW ID is used to indicate the gene that is deleted, but it is considered appropriate to use ECK ID rather than JW.

Response: To address the comment, we have revised Appendix Table S3.

4) Since there are many abbreviations, it would be easier to note them all together.

Response: We have added list of abbreviations to Appendix Text S3.

[References cited in this revision]

- Baba T, Ara T, Hasegawa M, Takai Y, Okumura Y, Baba M, Datsenko KA, Tomita M, Wanner BL, Mori H (2006) Construction of *Escherichia coli* K-12 in-frame, single-gene knockout mutants: the Keio collection. *Mol Syst Biol* 2: 2006.0008
- Dolan SK, Welch M (2018) The glyoxylate shunt, 60 years on. *Annu Rev Microbiol* 72: 309-330
- Ebrahim A, Lerman JA, Palsson BO, Hyduke DR (2013) COBRApy: COnstraints-Based Reconstruction and Analysis for Python. *BMC Syst Biol* 7: 74
- Fida MAFA, Ahmad T, Ntahobari M, 2021. Variance threshold as early screening to Boruta feature selection for intrusion detection system, 2021 13th International Conference on Information & Communication Technology and System (ICTS). pp. 46-50.
- Gudmundsson S, Thiele I (2010) Computationally efficient flux variability analysis. *BMC Bioinform* 11: 489
- Herrmann HA, Dyson BC, Vass L, Johnson GN, Schwartz J-M (2019) Flux sampling is a powerful tool to study metabolism under changing environmental conditions. *NPJ Syst Biol Appl* 5: 32
- Kim J, Copley SD (2007) Why metabolic enzymes are essential or nonessential for growth of *Escherichia coli* K12 on glucose. *Biochemistry* 46: 12501-12511
- Kohavi R, John GH (1997) Wrappers for feature subset selection. *Artif Intell* 97: 273-324
- Lundberg SM, Lee S-I (2017) A unified approach to interpreting model predictions. *Adv Neural Inf Process Syst* 30: 4768-4777
- Mahadevan R, Schilling CH (2003) The effects of alternate optimal solutions in constraint-based genome-scale metabolic models. *Metab Eng* 5: 264-276
- Monk JM, Charusanti P, Aziz RK, Lerman JA, Premyodhin N, Orth JD, Feist AM, Palsson B (2013) Genome-scale metabolic reconstructions of multiple *Escherichia coli* strains highlight strain-specific adaptations to nutritional environments. *Proc Natl Acad Sci USA* 110: 20338-20343
- Monk JM, Lloyd CJ, Brunk E, Mih N, Sastry A, King Z, Takeuchi R, Nomura W, Zhang Z, Mori H *et al* (2017) iML1515, a knowledgebase that computes *Escherichia coli* traits. *Nat Biotechnol* 35: 904-908
- Nie L, Ren Y, Schulz H (2008) Identification and characterization of *Escherichia coli* thioesterase III that functions in fatty acid β -oxidation. *Biochemistry* 47: 7744-7751
- Nygaard P, Smith JM (1993) Evidence for a novel glycinamide ribonucleotide transformylase in *Escherichia coli*. *J Bacteriol* 175: 3591-3597
- Otsu N (1979) A threshold selection method from gray-level histograms. *IEEE Trans Syst Man Cybern* 9: 62-66
- Sahu A, Blätke MA, Szymański JJ, Töpfer N (2021) Advances in flux balance analysis by integrating machine learning and mechanism-based models. *Comp Struct Biotechnol J* 19: 4626-4640
- Segrè D, Vitkup D, Church GM (2002) Analysis of optimality in natural and perturbed metabolic networks. *Proc Natl Acad Sci USA* 99: 15112-15117
- Tong M, French S, El Zahed SS, Ong WK, Karp PD, Brown ED (2020) Gene dispensability in *Escherichia coli* grown in thirty different carbon environments. *mBio* 11: e02259-02220
- Velur Selvamani RS, Telaar M, Friehs K, Flaschel E (2014) Antibiotic-free segregational plasmid stabilization in *Escherichia coli* owing to the knockout of triosephosphate isomerase (*tpiA*). *Microb Cell Fact* 13: 58
- Venkatesh B, Anuradha J (2019) A review of feature selection and its methods. *Cybern Inf Technol* 19: 3-26

- Yamamoto N, Nakahigashi K, Nakamichi T, Yoshino M, Takai Y, Touda Y, Furubayashi A, Kinjyo S, Dose H, Hasegawa M *et al* (2009) Update on the Keio collection of *Escherichia coli* single-gene deletion mutants. *Mol Syst Biol* 5: 335
- Yang JH, Wright SN, Hamblin M, McCloskey D, Alcantar MA, Schrübbers L, Lopatkin AJ, Satish S, Nili A, Palsson BO *et al* (2019) A white-box machine learning approach for revealing antibiotic mechanisms of action. *Cell* 177: 1649-1661.e1649

14th Dec 2023

Manuscript Number: MSB-2023-11953R

Title: Machine learning identifies key metabolic reactions in bacterial growth on different carbon sources

Dear Prof Yoon,

Thank you for the submission of your revised manuscript to Molecular Systems Biology. I am pleased to inform you that we will be able to accept your manuscript pending the following final amendments and appropriate response to reviewers:

- 1) Table EV1-EV2 should be renamed to Dataset EV1-EV2. Please also update their callouts in main manuscript text. The legends for these datasets should be included as a separate sheet in each Excel file.
- 2) Please format the Data availability section according to the example below:
The datasets and computer code produced in this study are available in the following databases:
 - Chip-Seq data: Gene Expression Omnibus GSE46748 (<https://www.ncbi.nlm.nih.gov/geo/query/acc.cgi?acc=GSE46748>)
 - Modeling computer scripts: GitHub (<https://github.com/SysBioChalmers/GECKO/releases/tag/v1.0>)
 - [data type]: [full name of the resource] [accession number/identifier] ([doi or URL or identifiers.org/DATABASE:ACCESSION])
- 3) Synopsis:
 - Synopsis image: The size and proportions of the synopsis image do not fit our requirements. Please upload the synopsis image as a high-resolution jpeg file 550 pixels wide x (250-400) pixels high.
 - Please check your synopsis text and image before submission with your revised manuscript. Please be aware that in the proof stage minor corrections only are allowed (e.g., typos).
- 4) As part of the EMBO Publications transparent editorial process initiative (see our Editorial at <http://embomolmed.embopress.org/content/2/9/329>), EMBO Molecular Medicine will publish online a Review Process File (RPF) to accompany accepted manuscripts. This file will be published in conjunction with your paper and will include the anonymous referee reports, your point-by-point response and all pertinent correspondence relating to the manuscript. Let us know whether you agree with the publication of the RPF and as here, if you want to remove or not any figures from it prior to publication. Please note that the Authors checklist will be published at the end of the RPF.
- 5) Please provide a point-by-point letter INCLUDING my comments as well as the reviewer's reports and your detailed responses (as Word file).

I look forward to reading a new revised version of your manuscript as soon as possible.

Yours sincerely,

Poonam Bheda, PhD
Scientific Editor
Molecular Systems Biology

Click on the link below to submit your revised paper.

Reviewer #1:

The authors have addressed most of my previous comments, and this version is much improved compared to the previous one. There is one point that remains unclear.

The authors claim that the discrepancy in detrimental reactions predicted by EN and MLP models can be attributed to the inherent characteristics of the output data and the feature selection methods. The important features were selected by SHAP interpretation, and I agree that EN and MLP may select different features as important. However, the statement "This can be attributed to the inherent characteristics of the output data and the feature selection methods." is not clear. Did they use different feature selection methods and output data for EN and MLP, or is it the same set of features? A clear statement justifying and discussing this methodological step would be needed.

Reviewer #2:

The authors have adequately addressed all the review comments.

Point-by-point response to referees' comments

The authors appreciate the editor and reviewers for the careful evaluation of the manuscript. Below, please find our detailed point-by-point responses to each of the concerns.

< Editor >

Thank you for the submission of your revised manuscript to Molecular Systems Biology. I am pleased to inform you that we will be able to accept your manuscript pending the following final amendments and appropriate response to reviewers:

1) Table EV1-EV2 should be renamed to Dataset EV1-EV2. Please also update their callouts in main manuscript text. The legends for these datasets should be included as a separate sheet in each Excel file.

Response: We have updated the tables and manuscript accordingly.

2) Please format the Data availability section according to the example below:

The datasets and computer code produced in this study are available in the following databases:

- Chip-Seq data: Gene Expression Omnibus GSE46748

(<https://www.ncbi.nlm.nih.gov/geo/query/acc.cgi?acc=GSE46748>)

- Modeling computer scripts: GitHub (<https://github.com/SysBioChalmers/GECKO/releases/tag/v1.0>)

- [data type]: [full name of the resource] [accession number/identifier] ([doi or URL or identifiers.org/DATABASE:ACCESSION])

Response: We have modified the Data availability section as follows.

All data, models, and codes used in this study are available on [GitHub](https://github.com/sybirg/xai_growth) (https://github.com/sybirg/xai_growth) with a DOI of [Zenodo](https://zenodo.org/records/10164986) (<https://zenodo.org/records/10164986>), along with information for replicating the presented results.

3) Synopsis:

- Synopsis image: The size and proportions of the synopsis image do not fit our requirements. Please upload the synopsis image as a high-resolution jpeg file 550 pixels wide x (250-400) pixels high.

Response: We have made the synopsis image file accordingly.

4) As part of the EMBO Publications transparent editorial process initiative (see our Editorial at <http://embomolmed.embopress.org/content/2/9/329>), EMBO Molecular Medicine will publish online a Review Process File (RPF) to accompany accepted manuscripts. This file will be published in conjunction with your paper and will include the anonymous referee reports, your point-by-point response and all pertinent correspondence relating to the manuscript. Let us know whether you agree with the publication of the RPF and as here, if you want to remove or not any figures from it prior to publication. Please note that the Authors checklist will be published at the end of the RPF.

Response: We agree with the publication of the RPF.

5) Please provide a point-by-point letter INCLUDING my comments as well as the reviewer's reports and your detailed responses (as Word file).

Response: In the accompanying "Response to Reviewer Comments" document, we have provided a point-by-point response to comments from the editor and reviewers.

< Reviewer 1 >

The authors have addressed most of my previous comments, and this version is much improved compared to the previous one. There is one point that remains unclear.

The authors claim that the discrepancy in detrimental reactions predicted by EN and MLP models can be attributed to the inherent characteristics of the output data and the feature selection methods. The important features were selected by SHAP interpretation, and I agree that EN and MLP may select different features as important. However, the statement "This can be attributed to the inherent characteristics of the output data and the feature selection methods." is not clear. Did they use different feature selection methods and output data for EN and MLP, or is it the same set of features? A clear statement justifying and discussing this methodological step would be needed.

Response: We thank the reviewer for the exhaustive review. To avoid possible misunderstandings, we have modified the related sentences as follows.

"The high discordance between the EN and MLP models was observed for predicting detrimental reactions, especially when seven reactions were predicted only by EN model to be detrimental under all carbon sources. This can be attributed to the inherent characteristics of the output data and the feature selection methods."

->

"Although the EN and MLP models were trained using the same dataset and feature selection method, they showed high discrepancy in predicting detrimental reactions (Fig. 2A). This was especially evident when the EN model alone identified seven reactions as detrimental across all carbon source conditions. This discrepancy can be attributed to the characteristics of the output data and the different methods employed by the EN and MLP models to calculate the feature importance."

< Reviewer 2 >

The authors have adequately addressed all the review comments.

Response: We thank the reviewer for the exhaustive review.

11th Jan 2024

Manuscript number: MSB-2023-11953RR

Title: Machine learning identifies key metabolic reactions in bacterial growth on different carbon sources

Dear Prof Yoon,

Thank you again for sending us your revised manuscript. We are now satisfied with the modifications made and I am pleased to inform you that your paper has been accepted for publication.

Yours sincerely,

Poonam Bheda, PhD
Scientific Editor
Molecular Systems Biology
